# Cortical population activity within a preserved neural manifold underlies multiple motor behaviors

Juan A. Gallego [1,2], Matthew G. Perich [3], Stephanie N. Naufel[3], Christian Ethier [4], Sara A. Solla[1,5] & Lee E. Miller [1,3,6]

Populations of cortical neurons flexibly perform different functions; for the primary motor cortex (M1) this means a rich repertoire of motor behaviors. We investigate the flexibility of M1 movement control by analyzing neural population activity during a variety of skilled wrist and reach-to-grasp tasks. We compare across tasks the neural modes that capture dominant neural covariance patterns during each task. While each task requires different patterns of muscle and single unit activity, we find unexpected similarities at the neural population level: the structure and activity of the neural modes is largely preserved across tasks. Furthermore, we find two sets of neural modes with task-independent activity that capture, respectively, generic temporal features of the set of tasks and a task-independent mapping onto muscle activity. This system of flexibly combined, well-preserved neural modes may underlie the ability of M1 to learn and generate a wide-ranging behavioral repertoire.

[1] Department of Physiology, Feinberg School of Medicine, Northwestern University, 303 E. Chicago Avenue, Chicago, IL 60611, USA. [2] Neural and Cognitive Engineering Group, Centre for Automation and Robotics CSIC-UPM, Ctra. Campo Real km 0.2 - La Poveda, 28500 Arganda del Rey, Spain. [3] Department of Biomedical Engineering, Northwestern University, 2145 Sheridan Road, Evanston, IL 60208, USA. [4] Département de Psychiatrie et Neurosciences, Université Laval, CERVO Research Center, 2601 Ch. de la Canardière, Québec, QC G1J 2G3, Canada. [5] Department of Physics and Astronomy, Northwestern University, Evanston, IL 60208, USA. [6] Department of Physical Medicine and Rehabilitation, Northwestern University, Chicago, IL 60611, USA. Correspondence and requests for materials should be addressed to J.A.G. (email: gallego.juanalvaro@gmail.com) or to L.E.M. (email: lm@northwestern.edu)

The generation of movement is crucial for survival. Whether seeking food, escaping a predator, or using tools to construct a shelter, motor behavior is arguably the ultimate purpose of the nervous system[1]. Primates, especially humans, have evolved an advanced cerebral cortex that allows for a rich repertoire of arm and hand movements. The activity patterns of neurons in the primary motor cortex (M1) during such movements are highly complex, and the mechanisms by which a given population of neurons controls varied behaviors remain unclear.

Historically, researchers have ascribed independent encoding functions to the activity of single neurons[2–4] or groups of neurons[5–8], and looked for correlations between neural activity and specific movement parameters. The hope was that neural activity would encode particular parameters in a largely invariant manner. However, considerable variability in the activity patterns across neurons[9,10] and instability of movement parameter encoding by single neurons across different conditions[4,11–18] have obscured the identification of simple underlying principles. An intriguing alternative is that the computations involved in generating movement are not based on the independent modulation of single neurons, but rather performed at the population level by networks of interconnected neurons[10,19–21] whose coordinated activity commands the muscles that cause the behavior[19,21–24]. In this view, correlates between single neuron activity and behavior are epiphenomenal[10,25–27] and yield only a limited and distorted view of the causal relation between M1 activity and behavior.

To study neural function at the population level, we describe neural activity in a high-dimensional neural space in which each axis represents the activity of one recorded neuron[21,28–30]. The dimensionality of this neural space is determined by the number of neurons that can be recorded simultaneously, until recently in the order of hundreds. This number is increasing rapidly[31], reaching $10^5$ and beyond[32]. This high dimensionality imposes practical and theoretical challenges, thus many recent studies have applied dimensionality reduction methods[28] to neural spaces for numerous brain areas. During typical laboratory tasks, neural activity has been found to explore only a limited, low-dimensional portion of the full neural space[21,33]. This sub-dimensional region, the neural manifold[21,29,30,34], reflects covariance patterns across the neural population activity (Fig. 1b). The neural manifold is spanned by a set of basis vectors, the neural modes[21] (Fig. 1b), patterns of neural covariance thought to arise from the network connectivity[35] (Fig. 1a, c). Evidence for the relation of the neural manifold to the underlying connectivity also arises from recent results on the difficulty of altering the orientation of the manifold voluntarily on a short learning timescale of only hours[29].

In the manifold view, the population activity captured by neural modes is the fundamental computational unit, while the activity of each single neuron is simply a one-dimensional (1-D) projection of this population activity[21,36,37] (Fig. 1c). Neurons thus likely provide only randomly oriented, low-dimensional glimpses[36] of the overall neural picture, the neural analog of blind men examining an elephant. Recent experimental and theoretical evidence is consistent with the view that neural function may be built upon the coordinated activation of these neural modes[19–21,38]. Almost all of these experiments have focused on a single task or behavior[19,24,34,38–41], which makes it difficult to determine the extent to which the identified manifold truly captures fundamental building blocks of neural activity (an exception is ref. [42], a comparison of rodent reaching and walking). The open question is that of potential similarities among manifolds identified in the same neural population during a variety of skilled motor tasks.

Here, we hypothesize that the motor cortex generates varied skilled behaviors through the flexible activation of different

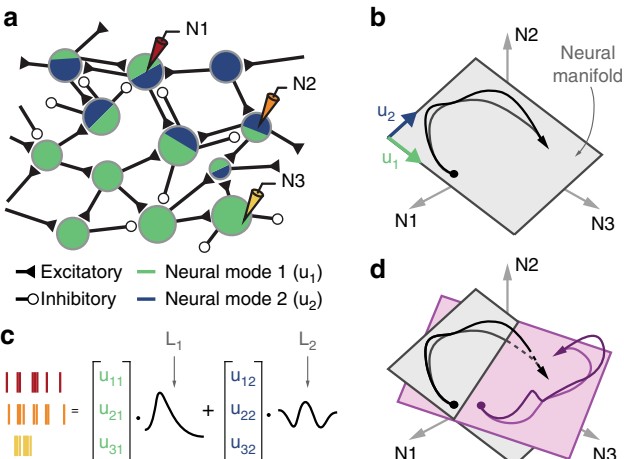

**Fig. 1** Hypothesis: varied motor behaviors are caused by the flexible activation of combinations of neural modes. **a** The connectivity of the cortical network results in neural modes whose combined activity explains the specific activity of individual neurons. **b** The neural space for the three neurons recorded in **a**. The time-dependent population activity is represented by a trajectory (in black, arrow indicates time direction) mostly confined to a two-dimensional neural manifold (gray plane) spanned by two neural modes (green $u_1$ and blue $u_2$ basis vectors). **c** The time-dependent activity of each recorded neuron is a weighted combination of the latent activities $L_1$ and $L_2$, each the time-dependent activation of the corresponding neural mode. **d** Do neural manifolds for different tasks (shown in gray and light purple) have similar orientation? Are the latent activities for the two tasks (shown in black and purple) similar? These are the two critical questions to test our hypothesis

combinations of neural modes that are not fully task-specific. To examine this hypothesis, we recorded neural population activity from M1 during several skilled arm and hand tasks—defined here as those that require independent control of individual joints[43]—and identified the corresponding neural modes. Despite widespread differences in both neural activity and motor output, the orientation of the corresponding manifolds within the neural space was unexpectedly well-preserved across tasks. We then studied the time-dependent activation of the neural modes, the latent activity[21,28,34,44] that characterizes the neural population dynamics within the manifold. We found significant similarities for the different tasks, with the activity of one set of preserved neural modes being strongly predictive of muscle activity patterns (EMG) across tasks. Our observations stand in contrast to previous studies that tried to identify stable behavioral correlates of single neuron activity and found instead that these relationships were quite labile[10,11,13,15,18,45–47]. The presence of neural modes in other brain areas, like frontal, prefrontal, parietal, visual, auditory, and olfactory cortices (see refs. [21,28] for recent reviews) suggests that the activation of flexible combinations of neural modes may be the general mechanism underlying the population dynamics associated with neural computation.

## Results

**Hypothesis**. We hypothesized that skilled motor behaviors are generated by the flexible activation of different combinations of well-preserved neural modes, rather than the independent modulation of single neurons. Consider a simple three-neuron example (Fig. 1b). The population activity during movement traces a trajectory that could in principle explore all regions of the neural space. In practice, correlations (covariations) between

neurons constrain the possible population patterns[29] and thus the region of neural space actually explored by the population dynamics. We use a dimensionality reduction technique, such as principal component analysis (PCA)[28,30,34] to identify two dominant neural modes for this trajectory ($u_1$ and $u_2$ in Fig. 1b). These two modes span a two-dimensional (2-D) neural manifold[21,29,30], the plane where the trajectory is largely confined (Fig. 1b).

Based on the presumed relationship between network connectivity and the neural manifold[21,29,35], we hypothesized that varied skilled arm and hand tasks would involve similar neural modes. If correct, this hypothesis would have two implications. First, neural manifolds for different tasks should be relatively well aligned, a prediction we can test using principal angles[48]. As an example, consider the 2-D manifolds in Fig. 1d. Since they intersect, the smallest principal angle is actually zero. The second principal angle is the small angle between the two manifolds. These small principal angles quantify similar orientation and indicate that much of the neural covariance structure is preserved across these two tasks. Second, we would expect similarities in the dynamic activation of these neural modes[49], the latent activity; these similarities would arise from generic activity patterns intrinsic to the network[26,50] and/or from similarities in motor output across tasks. Demixed PCA (dPCA)[51] and canonical correlation analysis (CCA)[22,52] are useful tools for comparing latent activities.

**Behavioral tasks and neural recordings**. To study M1 activity, we recorded data using 96-channel microelectrode arrays chronically implanted in the hand area of M1 of three rhesus macaque monkeys (Monkeys C, T, and J). In each session, the monkeys performed one of two sets of motor tasks (Methods). The first set comprised several wrist tasks used to dissociate kinematics and forces needed to acquire a given target, a strategy used in many prior single neurons studies[2–5,16,53]. This set of tasks included 1-D isometric, unloaded, and elastic-loaded movements (Monkeys C and J); Monkey J also performed a 2-D version of the isometric task[12]. The other set (Monkeys T and C) included a power grip task[54,55], and a task that required a ball to be grasped, transported, and dropped[55]. One task of each set is illustrated in Fig. 2. In the 1-D isometric wrist task (Fig. 2a), the monkeys controlled cursor movements to specific targets through the torque exerted at the wrist (Fig. 2b; see Fig. 2c for the torque trajectories during the 2-D isometric task, and Fig. 2d for the torques and velocities for all wrist tasks). In the power grip task (Fig. 2f), the monkeys reached for and grasped a pneumatic tube, squeezing it to achieve a target force (Fig. 2g). During each task, we identified neural units through threshold crossings on each electrode. Across all tasks, sessions, and monkeys, we detected $65.9 \pm 16.9$ (mean ± s.d.; range, 45–91) units with waveforms that were stable throughout a given session. Each task was associated with distinct patterns of muscle (Supplementary Fig. 1) and neural activity (Fig. 2e, h, Supplementary Fig. 2). Complex changes in the activity of individual units across tasks resulted in relatively low correlations in the activity of individual units across tasks (Fig. 2i).

For each task, we used PCA to identify leading neural modes spanning a 12-dimensional (12-D) manifold[21,28,30,34] (Methods). Activity confined to these 12-D manifolds accounted for $73.4 \pm 6.5\%$ of the variance for all tasks, across all datasets (Supplementary Fig. 3a, c). This dimensionality is comparable to that reported for populations of neurons in the arm area of M1 during reaching[29,38] and reach-to-grasp tasks[40,56]. Notably, most units contributed to all neural modes; each mode captured a population-wide activity pattern (Supplementary Fig. 3b, d).

Neural modes and their activity were robust against the particular choice of randomly sampled units: both were remarkably well preserved even if computed from only 60% of the recorded units (Supplementary Fig. 3e, f). The low dimensionality of these manifolds and their robustness against eliminated units indicate that they can be reliably estimated from the activity of ~100 recorded units, even though these are only a very small fraction of the participating population. These observations support the view that the orientation of the neural manifold and the activity within it do not depend on specific recorded units[57] and are not an artifact of their currently inevitable undersampling[33,37].

**Comparison of neural modes across motor tasks**. If M1 does indeed generate movement through flexible combinations of well-preserved neural modes, the neural manifolds corresponding to different tasks should be similarly oriented. To test this conjecture, we computed the 12 principal angles[48] between the 12-D manifolds for all pairs of tasks during each session (Methods). Our hypothesis predicts that these angles will be small. In contrast, if M1 recruited neurons in arbitrary combinations rather than as part of stable neural modes, the covariance structure would change across tasks and the corresponding manifolds would not be similarly oriented. To provide an intuition for the value of the 12 principal angles between two 12-D manifolds within a high-dimensional neural space, we computed the distribution of principal angles obtained from a null hypothesis generated with the tensor maximum entropy method[27]. For each comparison to experimental results, we generated surrogate neural activity patterns that preserved the covariance of the original data over time and across targets, but not across neurons, since this was the variable whose stability we tested. We computed the distributions of the 12 principal angles between the 12-D manifolds for the surrogate data (Methods, Supplementary Fig. 4a,b), and used them to set a conservative threshold ($P < 0.001$) below which the principal angles between manifolds for different tasks were considered significantly small.

Representative individual sessions in Fig. 3a show the leading principal angles between pairs of task-specific manifolds; these were far smaller than the surrogate threshold (dashed lines in the figure), with only the final non-leading two angles comparable to the surrogate. When pooling across all monkeys, sessions, and task comparisons, most principal angles were well below this threshold (Fig. 3b; all datasets in Supplementary Fig. 4c). As an example, the three leading principal angles averaged $8.4 \pm 2.3°$, $11.3 \pm 2.9°$, and $15.1 \pm 4.6°$, while the corresponding surrogate thresholds were $31.0 \pm 7.4°$, $38.9 \pm 6.9°$, and $44.6 \pm 6.4°$. This result held for a broad range of manifold dimensionalities (Supplementary Fig. 5a shows from 8 to 15), and when using single units instead of multi-units to compute the manifolds (Supplementary Fig. 4e).

To better interpret this degree of similarity across manifolds corresponding to different tasks, we projected the data from one task onto the manifold of another task, computed the neural variance accounted for (VAF) by this 12-D manifold (across-task VAF), and compared it to the VAF when the same data were projected onto its original 12-D manifold (within-task VAF)[58]. The across-task VAF was as large as $83.0 \pm 6.6\%$ of the within-task VAF (Fig. 3c), a significantly large result when compared to the VAF ratios obtained by projecting data on a randomly oriented manifold with maximal VAF ($P{\sim}0$, two-sided Wilcoxon rank sum test; see Methods).

The strong similarity in the orientation of manifolds corresponding to different skilled arm and hand tasks indicates that the structure of neural modes activated by these tasks is well preserved, as predicted by our hypothesis. The structure of the

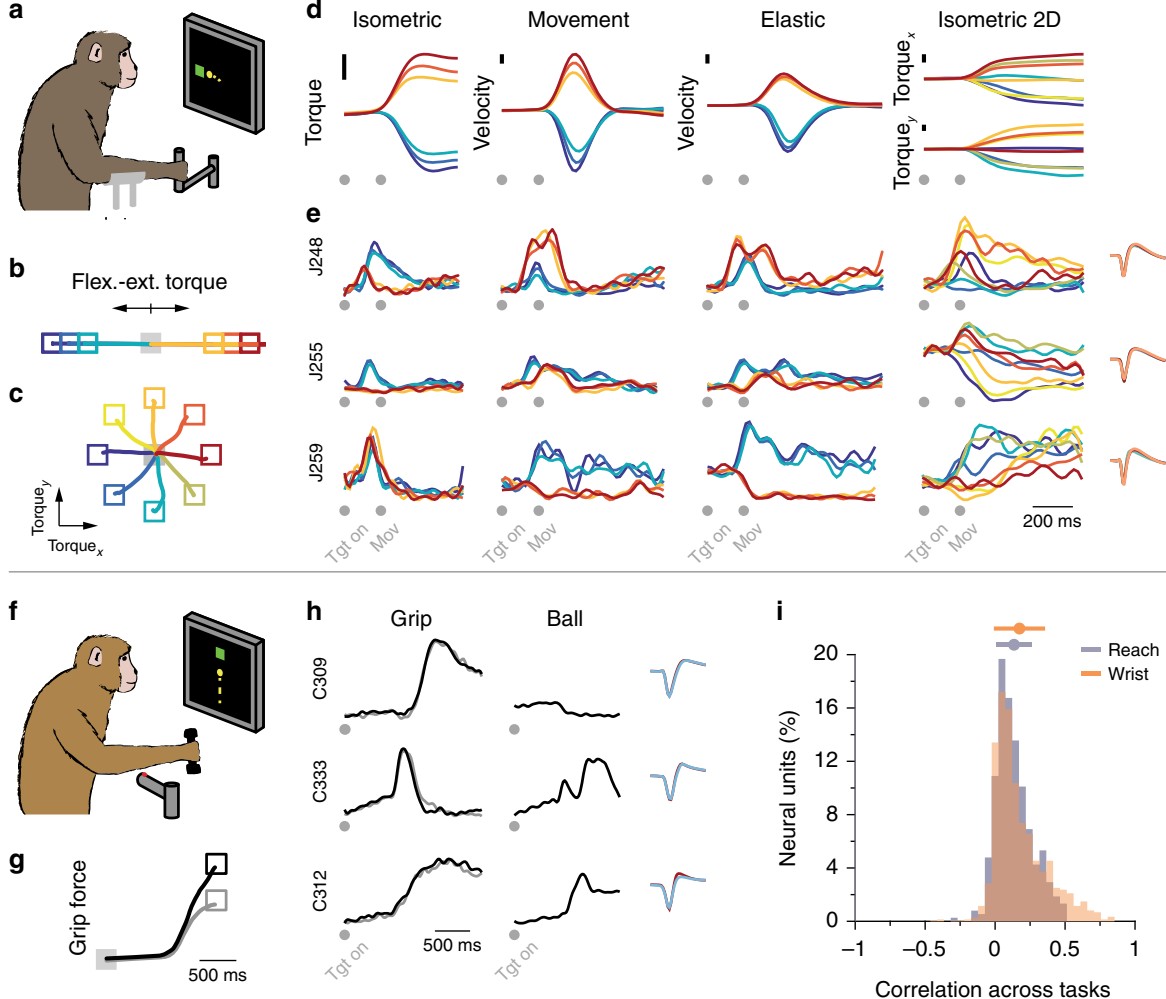

**Fig. 2** Tasks and recordings. **a** The wrist isometric center-out task. Torque trajectories to acquire each target (colored squares) during the 1-D isometric (**b**) and 2-D isometric (**c**) tasks. **d** Exerted wrist torque or movement to acquire each target. **e** Three neural units illustrate the variety of activity patterns across units, and how these patterns change across tasks. Right inset: overlaid action potential waveform for each unit across tasks; each task in a different color. Note the excellent overlap. Data for **b**–**e** are from Monkey J, averaged over all trials in one session, and colored according to target location (**b**, **c**; see also Supplementary Fig. 2). **f** The power grip task. **g** Grasp force trajectories to acquire each target (black or gray squares). **h** Three neural units illustrate the variety of activity patterns across units for the grip task, and how these patterns change in a complex manner for the ball task. Right inset: overlaid action potential waveform for each unit across both tasks; each task in a different color. Note the excellent overlap. Data for **g** and **h** are from Monkey C, averaged over all trials in one session, and colored according to target (**g**). **i** Correlations $r$ between the activity pattern of each neural unit across each pair of tasks, pooled over all units, task comparisons, sessions, and monkeys for the wrist and reach-to-grasp datasets separately; top error bar: correlation mean ± s.d.; for the wrist datasets, $\bar{r} = 0.15 \pm 0.19$, $n = 1476$; for the reach-to-grasp datasets, $\bar{r} = 0.11 \pm 0.13$, $n = 376$

population covariance patterns was thus strongly conserved despite the complex changes in the activity of individual units across tasks.

**Task-specific and task-independent latent activity**. Given the significant degree of task-independence in the orientation of the neural manifold, we ask: how do M1 populations generate motor output appropriate to distinct tasks in the face of this largely preserved neural covariance? One intriguing possibility is that the brain exploits the flexible activation of a fairly stable set of neural modes to generate task-specific motor commands.

To investigate this possibility, we used dPCA[51] to find a single neural manifold for all tasks in a given session. We relied on dPCA's ability to identify neural modes whose activity covaries with specific behavioral parameters to assess the task-specificity of the corresponding latent activity (Methods; Supplementary Fig. 6a). Specifically, we looked for four different types of neural modes. The activity of two types of modes was task-independent:

the activity of time-related modes depended only on time (i.e., progression through a trial) and was unrelated to specific details of the motor output, and the activity of target-related modes depended on the exerted movement or torque, as determined by the location of the target. The activity of two types of modes was task-dependent: the activity of task-related modes depended only on the task being performed, and the activity of task/target modes related to both task and the direction of the exerted movement or torque. We chose these covariates to dissociate latent activity features intrinsic to the neural dynamics from those related to potential commonalities in motor output imposed by task and target similarities. We confined this analysis to the wrist datasets (1-D isometric, movement, and elastic-loaded movement tasks for Monkeys J and C, and also 2-D isometric for Monkey J; Methods), to analyze both task and target dependence of the latent activity. CCA[22,52], an alternative approach that we could also apply to the reach-to-grasp datasets, yielded similar results (Methods; Supplementary Fig. 7).

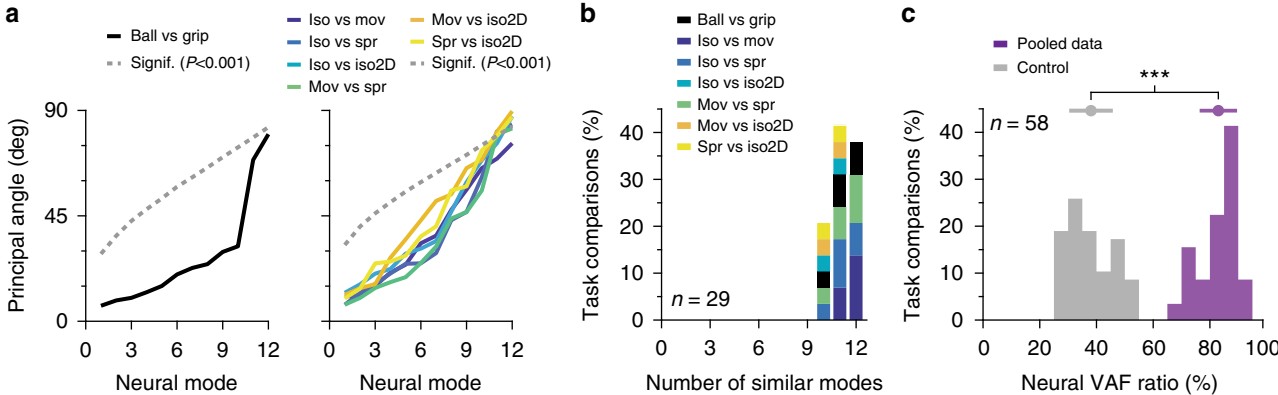

**Fig. 3** Comparison between neural manifolds for two different tasks. **a** Principal angles for one session of reaching and grasping tasks from Monkey T (left) and for one session of wrist tasks from Monkey J (right). Each pairwise comparison is shown as one colored trace (see legend). Leading principal angles were far below the $P < 0.001$ significance level (dashed gray line), indicating significant similarities in the structure of the neural modes across tasks. For simplicity, only the most stringent significance threshold is shown for all pairwise comparisons across wrist tasks (right panel). **b** Number of neural modes for which all principal angles were significantly small across all monkeys, sessions, and task pairs ($P < 0.001$; surrogate threshold obtained with the tensor maximum entropy method). **c** Ratio of the variance accounted for (VAF) when projecting the neural data from one task onto the neural manifold from another task, to the VAF when projecting the same data onto its corresponding manifold (purple); data pooled over all monkeys, sessions, and task pairs. We compared this distribution to a control based on the 99.9th percentile of the distribution of VAF ratios when projecting neural data onto randomly generated manifolds (gray); the *** denote $P \sim 0$ (two-sided Wilcoxon rank sum test)

Given the small principal angles between manifolds corresponding to different tasks (Fig. 3), dPCA unsurprisingly found a single 12-D neural manifold that captured most of the population variance across all tasks ($95.5 \pm 0.5\%$ of the VAF by 12-D regular PCA; Supplementary Fig. 6b, f). Each of the dPCA neural modes that spanned this manifold covaried almost exclusively with one of the chosen behavioral parameters (Fig. 4a; additional example in Supplementary Fig. 6c). Although the activity of individual neural units was strongly task-dependent, as evidenced by their low correlations across pairs of tasks (Fig. 2i), neural modes that shared task-independent activity captured more than half of the total neural variance (65.5% for Monkey C, 59.0% for Monkey J; see Fig. 4c). As was the case for the principal angle analysis, these results held for a broad range of manifold dimensionalities (Supplementary Fig. 5b).

Figure 4b shows the latent activity corresponding to the eight leading dPCs for each of the 24 task-target combinations (four tasks × six targets; see Fig. 4d for target organization) for one representative session (Supplementary Fig. 6d, e show one example session from another monkey). The top row in Fig. 4b shows neural modes whose activity was virtually identical for all targets and tasks; a striking similarity given the different time courses of the corresponding motor outputs (Fig. 2d, Supplementary Fig. 1). The second row in Fig. 4b shows additional task-independent activity for target-related modes that capture the direction of action, indicated by the sign of the corresponding velocity or torque. This activity separated targets requiring wrist extension from those requiring wrist flexion, regardless of the specifics of the task.

The third row in Fig. 4b shows neural modes with task-specific activity. For example, mode 3 displays a task-related offset that distinguishes the 1-D tasks from the 2-D task. This offset, present well before the movement is initiated, presumably represents some aspect of movement preparation[22,24,38,41,58] that is task but not target specific. Activity of the modes in the fourth row in Fig. 4b covaried jointly with task and target, following complex patterns. Analysis of these dPCs shows that not only was the structure of the neural modes preserved across different tasks, but that the activity of a subset of them, which explained as much as ~60% of the total neural variance (Fig. 4c), was also stable. Thus, M1 may generate different behaviors through the flexible activation of different combinations of a well-preserved set of neural modes, rather than through the modulation of the activity of individual neurons (Supplementary Fig. 7g).

**Do motor output similarities explain manifold similarities?**. We investigated whether the observed similarities in neural mode structure and activity might be expected based simply on similarities in the associated motor output, characterized here by muscle activity (EMG). We first tested whether the covariance of the EMG patterns was as well-preserved across tasks as the covariance of the neural activity. We used PCA to identify the EMG manifold associated with each task. These manifolds are spanned by leading muscle covariance patterns, the EMG modes (analogous to muscle synergies[59,60]). We then computed the principal angles between EMG manifolds for different tasks. The EMG manifolds were much less well aligned across tasks than the corresponding neural manifolds (Fig. 5a). Therefore, the observed across-task similarities in the orientation of neural manifolds are not simply a reflection of a similarly well-preserved orientation in the EMG manifolds associated with the motor output.

Next, we compared the similarity in latent activity to the similarities in EMGs. We used CCA[52], a method for comparing multi-dimensional time series[22] (Methods). This analysis yields as many canonical correlations (CC) as variables being compared; each CC ranges from 0 to 1, with 1 indicating perfect correlation. The leading CCs between latent activities across tasks were considerably higher than those between EMGs (Fig. 5b, c); the substantial similarities in latent activity cannot be trivially explained by comparable similarities in muscle activity. This observation is further supported by the contrasting result obtained with a computational model that simulates M1 activity corresponding to observed EMGs (Methods; Fig. 5d). We simulated neural activity from the EMGs for each pair of tasks, and observed that for any given number of neural modes, the simulated activity captured a larger fraction of neural variance than the actual data (Fig. 5e). For the simulated data, the CCs across tasks were over two times lower than those for the actual neural data ($2.3 \pm 3.9$; see Fig. 5f for a comparison of the across-task latent activity CCs from the real and simulated neural data; $P \sim 0$, two-sided Wilcoxon rank sum test). Together, these two

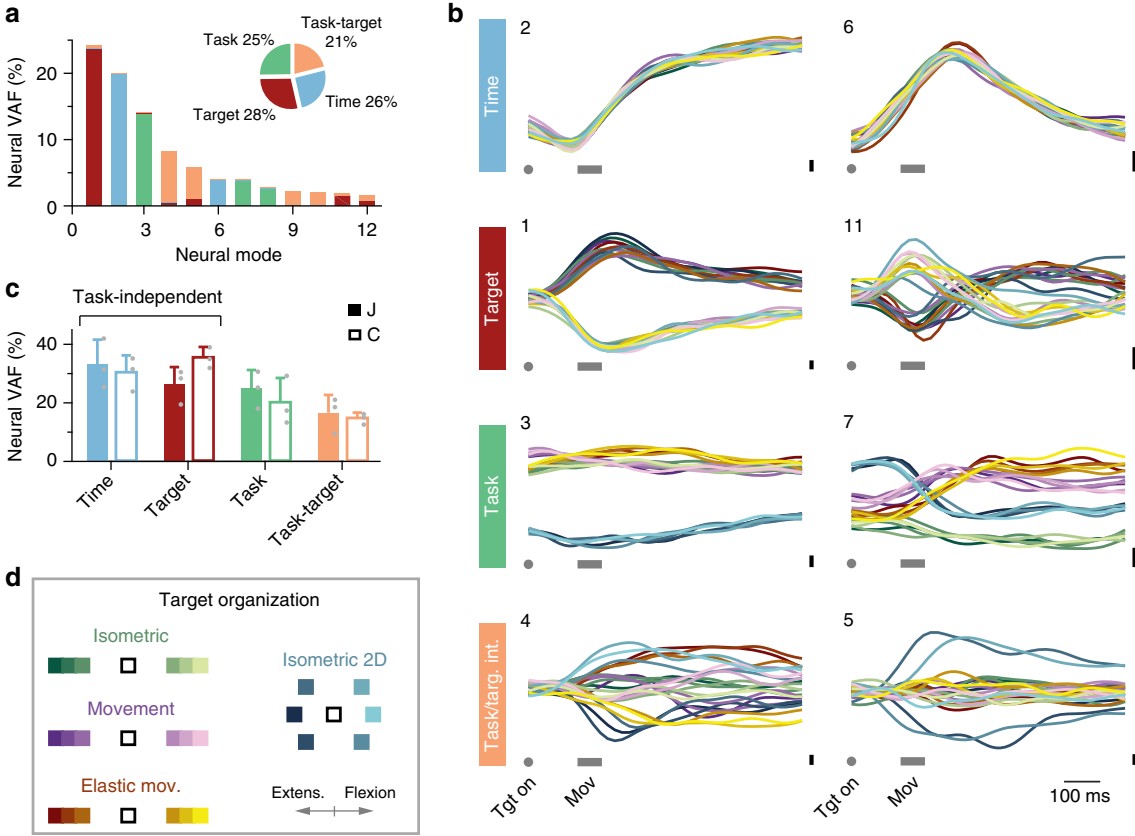

**Fig. 4** Task-specific and task-independent activity of dPCA modes. **a** Neural variance accounted for (VAF) by each neural mode, and its relation to behavioral parameters for one example session from Monkey J. Inset: amount of VAF associated with each behavioral parameter across all 12 modes. **b** Activity of eight dPCA neural modes, grouped in four sets based on the behavioral parameter they are most strongly associated with. The number on the top left of each panel indicates the ranking of that neural mode in terms of VAF, as in **a**. Each row corresponds to one behavioral parameter (vertical labels on the left). Each panel has 24 traces, corresponding to each of 24 task/target combinations; color code shown in **d**. **c** Total VAF for each behavioral parameter, averaged over sessions for Monkeys J and C. Bars: mean + s.d.; gray circles: individual sessions. **d** Target locations for each wrist task, color-coded for each task/target combination in **b**. Tasks are color-coded; extension targets are shown in dark colors and flexion targets in light colors

results indicate that neural activity synthesized from EMGs is both lower dimensional and less dynamically correlated across tasks than the actual neural activity.

**From neural modes to muscle commands**. For any given task, EMGs can be reasonably well predicted based on the activity of either a group of neurons[23,61] or the neural modes[19,22,24,26]. Using dPCA, we identified a set of target-related neural modes whose largely task-independent activity was related to the motor output. Their latent activity separated wrist flexion from wrist extension (Fig. 4b, modes 1 and 11; Supplementary Fig. 8e, modes 1 and 7), which suggests that these neural modes may relate to a target-dependent but task-independent component of EMG activity. To test for this possibility, we built a standard Wiener cascade decoder[55,61] for all wrist tasks in a given session, using only the two leading target-related dPCs as EMGs predictors (Methods). If the task-independent aspect of these modes' activity is relevant to the EMGs, these decoders should yield good predictions.

As exemplified in Fig. 6a, these decoders made EMG predictions that were >50% as accurate as those made with decoders based on all 12 neural modes as inputs (normalized $R^2$ across all muscles, tasks, and monkeys: 0.52 ± 0.32; see Methods). Moreover, these decoders far outperformed all other decoders based on two neural modes related to any of the other three behavioral parameters: time, task, and task/target (Fig. 6b; by paired two-sided Wilcoxon rank sum test, $P \sim 0$ for all three

comparisons). Finally, the ability to predict EMGs from the target-related neural modes cannot be attributed simply to the amount of neural variance of these modes explained (Fig. 6c); time-related neural modes explained more variance than the target-related neural modes, yet their EMG predictions were worse by a factor of ~2.5. The target-related neural modes were good predictors of the EMGs because a large fraction of the EMG variance was target-related and task-independent (Supplementary Fig. 8). Thus, the target-related neural modes identified directions within the manifold that captured a task-independent contribution of neural activity onto muscle activity.

## Discussion

Motor cortical population activity during a variety of skilled motor tasks is well described by a small number of neural modes, a set of population activity patterns[19–21,24,38,40,41,56,62] that span the neural manifold. These neural modes are putatively related to the connectivity of the network[29,35,49], and their activity is a good predictor of behavior[19,24,39–42,62,63]. Here we reported the first comparison of neural manifolds associated with a variety of skilled arm and hand tasks. Our results suggest that different behaviors may be generated by the activation of flexible combinations of a set of well-preserved neural modes[20,21], since: (1) despite the distinct patterns of muscle and neural unit activity associated with these tasks, the structure of the neural modes was largely preserved; (2) based on their activity, we could divide these modes into task-specific and task-independent sets, with the

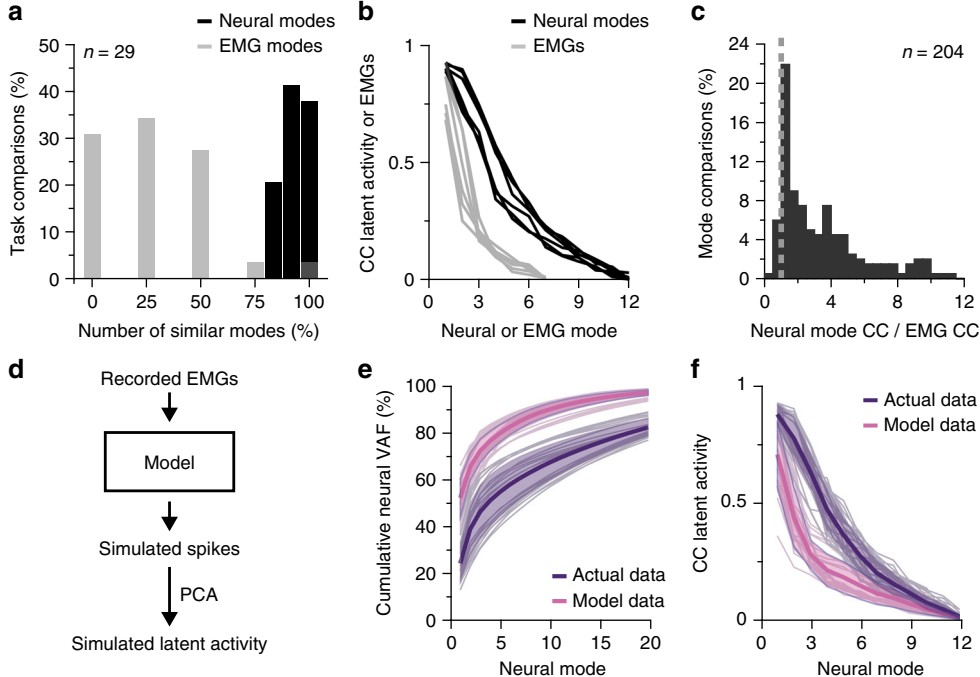

**Fig. 5** Similarities in latent neural activity are not accounted for by similarities in motor output. **a** Comparison between the percentage of neural modes and EMG modes that are preserved across tasks using principal angles; histogram shows the percentage of principal angles smaller than the $P <$ 0.001 significance threshold. Neural manifold: 12-D, EMG manifold: 4-D; data pooled over all tasks, sessions, and monkeys. **b** Example comparison of canonical correlations (CCs) for latent neural activity (black) and EMGs (gray); all pairwise comparisons across wrist tasks for one session from Monkey J. Each of the six pairwise comparisons for four wrist tasks is shown as a separate trace. **c** Histogram of the ratio of the pairwise CCs for latent neural activities to the pairwise CCs for EMGs across tasks (as shown in **b**). Data pooled over all tasks, wrist sessions, and monkeys. Most ratios are quite above one (dashed vertical line). **d** Neural activity was simulated as weighted linear combinations of the recorded EMGs; the neural manifold and latent activity were then computed for the simulated data. **e** Variance accounted for (VAF) as function of the number of neural modes for the simulated and the actual neural data. Data pooled over all monkeys, wrist sessions, and tasks; thick trace and colored strip: mean ± s.d.; colored traces: individual task comparisons. **f** The pairwise CCs were significantly smaller for the simulated than for the actual neural data (same data as in **e**)

latter able to explain ~60% of the total variance across different wrist tasks; and (3) the set of neural modes with task-independent activity included some that mapped consistently onto EMGs. These results provide new insight into how movement is generated by the motor cortex.

Neural population activity, and thus latent activity, potentially includes intrinsic dynamics and response to inputs in addition to outputs[16,21,22,26]. Our dPCA analysis identified time-related neural modes whose activity was virtually the same regardless of the task and even the action required (top row in Fig. 4b). These modes are thus unlikely to reflect inputs or outputs; instead, they probably capture generic temporal features[19,22,26], a common temporal evolution of the network during the set of tasks used here. Their activity may reflect the transition from a planning state to a movement state; the activity of the leading time-related mode has been found to predict the reaction time of reaching movements with great accuracy[50]. Time-related neural modes may also support the generation of robust motor commands[26], or the state-dependent switching from active postural control to movement control[11,64].

However, similarities in latent activity across tasks are not fully explained by time-related modes. We also found target-related modes that had a task-independent mapping to muscle activity (Figs. 4 and 6), a mechanism that might help simplify limb control as well as motor adaptation[41]. For example, adaptation to a force field may be accomplished using alternative latent activity within the same manifold that controls the unperturbed movement[41]. This adaptation mechanism would be simpler than using a force-field-specific mapping to muscle commands, requiring not

only alternative latent activity but also an alternative mapping onto muscle activity.

Our finding of a task-independent mapping of M1 activity onto EMGs might be due to the similarity in motor action for similarly arranged targets in the wrist tasks considered here (Fig. 4d). Although the EMGs for a given target were quite different across the wrist tasks (Supplementary Fig. 1), the similarity in target arrangements resulted in a target-related but task-independent EMG mode that separated wrist flexion and extension (Supplementary Fig. 8). This preserved M1 to EMG mapping was found across isometric and movement tasks that require quite different muscle (Supplementary Fig. 1) and neural unit activity patterns (Fig. 2, Supplementary Fig. 2).

This task-independent M1 to EMG mapping is in contrast to the behavior-specific mapping found in mice when forelimb population activity during reaching was compared to that during treadmill walking[42]. The corresponding manifolds of these two tasks were orthogonal[42], with none of the similarities in orientation and activity found here. This lack of similarity is perhaps not surprising, as M1 is less directly involved in the control of treadmill walking than of reaching[42,65]. In contrast, we expect M1 neurons to be directly involved in the full range of skilled reach-to-grasp and wrist/hand motor behaviors[13,53,54,66,67] of the type studied here. We were unable to compare the reach-to-grasp tasks to the wrist tasks because they were studied in different monkeys or using different microelectrode arrays (Methods), but we conjecture that when compared, the neural modes for these tasks would exhibit many of the similarities reported here.

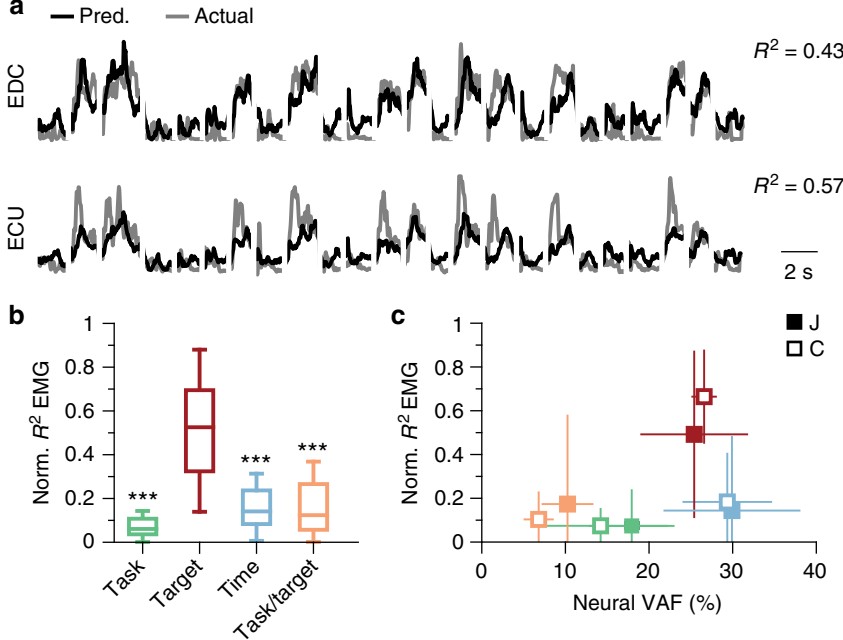

**Fig. 6** Neural modes with task-independent activity as predictors to muscle activity. **a** Example EMG predictions for two muscles during 24 trials of the 1-D isometric task for one of the sessions from Monkey J. Predictions were based on the two leading target-related neural modes. **b** Normalized $R^2$ of the EMG predictions for four different decoders, each based on the two leading neural modes most strongly related to the corresponding dPCA behavioral parameter. Performance was averaged across all muscles, wrist tasks, and monkeys. Boxplots show median: center line; interquartile range: box; and data range (0.5 × interquartile range): whiskers. The *** denote $P{\sim}0$ (two-sided Wilcoxon rank sum test). As reference, the $R^2$ of the decoders based on all 12 dPCA modes was 0.53 ± 0.21. **c** Normalized $R^2$ of the EMG predictions vs. the neural variance accounted for (VAF) by each of the four sets of neural modes identified with dPCA. Data averaged across all sessions and muscles for each monkey separately (legend). Squares: median ± s.d.; color code as in **b**. Same data as in **b** and Fig. 4c

Increasing evidence, primarily from sensory systems[21,35], suggests that the neural covariation patterns captured by the neural modes may arise from the underlying network connectivity. In M1, the most compelling evidence for this view comes from a brain–computer interface study in which monkeys trained to produce accurate cursor movements were subjected to decoder perturbations[29]. Perturbations outside the existing manifold proved to be considerably more challenging to learn than perturbations within the manifold, which simply required activating the existing neural modes in novel combinations[29]. In view of this reported difference between within-manifold and outside-manifold learning, our results on the similar orientation of manifolds associated with different skilled motor tasks suggest that learning one of those tasks would significantly facilitate the acquisition of any of the others, a facilitation not expected between tasks with orthogonal manifolds.

Although our experiments did not directly probe the presumed relationship between neural modes and network connectivity, they do provide additional indirect evidence. First, if the covariance patterns captured by the neural modes were mere by-products of task constrains, it is unlikely that they would be so well-preserved across tasks (Fig. 3). Second, the activity of two subsets of these well-preserved neural modes was largely correlated across wrist tasks (Fig. 4); a qualitatively similar observation held for the reach-to-grasp tasks (Supplementary Fig. 7). These structural and dynamic similarities seem unlikely if the neural modes did not reflect a stable property of the underlying neural network.

Prior studies have tried to explain how M1 controls movement by looking for a fixed relationship between single neuron activity and behavioral parameters, to elucidate what single neurons encode or represent. The differences in activity across M1

neurons hints at the challenge of this approach[9,10,13,53]. Indeed, few individual neurons have simple, invariant tuning to a single behavioral parameter[4,10,11,14–16,25]. Instead, they typically correlate with a variety of behavioral parameters, including hand position, joint kinematics, end point force, and muscle activity, among others (for reviews, see refs.[10,15]). These observations have led to varied interpretations, for example, that different parameters are encoded by distinct subclasses of neurons[4,14,15], or that single neurons simultaneously encode multiple parameters[18,45,46,68].

Many experiments, beginning with the studies of Evarts[2], Humphrey[5], and Thach[3], have attempted to dissociate these confounded behavioral parameters by using different limb postures, workspaces, or external loads[4,12–14,16–18,53,54]. The ultimate goal was to identify fundamental representations that are stable against such manipulations. Cheney and Fetz found that the relationship between the activity of cortico-motoneuronal cells and the exerted torque changes during isometric and elastic-loaded movement wrist tasks[53]. Others investigated the stability of single neuron representations across different postures during isometric and movement wrist tasks, and found that only a limited fraction of M1 neurons change their activity in a way that parallels the changes in task kinematics or EMGs[4,12]. Similar manipulations have been used during reach-to-grasp tasks, to find that the relationship between the activity of corticomotoneuronal cells and the EMGs of their target muscles changed between power grip and precision grip tasks[54]. The relationship between the activity of single M1 cells and the exerted torque was also found to change across task conditions[13]. These studies highlight that single neuron representations of movement parameters are quite labile.

In contrast with this representational view, other groups have adopted the view that the dominant function of M1 is that of a controller that causes movement; in this view, the neural representations described above are merely a by-product of this role[15,20,21,25,26]. Among these studies, those that have adopted the population approach of the neural manifold have so far focused on single skilled motor tasks. Our experiments and analyses were designed to test whether the neural manifolds for different tasks have similar orientations and activity. The observed commonalities provide strong evidence that the neural manifolds capture fundamental principles of the cortical control of movement.

Neural modes are not restricted to the motor cortices (for reviews, see refs. [21,28]). This observation raises an intriguing question: do populations of neurons in other cortical areas also use flexibly activated neural modes to perform their varied functions? The application of dPCA to neural recordings during sensory discrimination has revealed manifolds with stimulus-related, decision-related, and time-related modes, in both monkey prefrontal and rat olfactory cortex[51]. Similarly, during a working memory task, population activity in prefrontal cortex was associated with a manifold spanned by modes linearly related to memory storage and stimulus response[69]. Thus, population activity in multiple cortical areas is associated with neural manifolds whose modes relate strongly to task-relevant parameters. The similarity between these results and the ones reported here for M1 suggests a general mechanism by which the brain could flexibly perform varied functions.

In summary, we have shown that manifolds associated with different skilled motor tasks have similar orientation. It is the covariance among units that is well-preserved across tasks, as opposed to the activity of independently modulated M1 neurons. Moreover, in contrast with the highly varied patterns of muscle and neural unit activity, the activity of two sets of neural modes was stable across behaviors. One of these sets captured generic temporal features of the tasks, unrelated to the details of the motor output, and the other provided inputs to a stable task-independent mapping onto muscle activity. These results support the view that motor cortex may control different motor behaviors through the flexible activation of different combinations of neural modes, which themselves may arise from network connectivity. We further suggest that a similar mechanism may underlie the ability of other cortical areas to perform varied non-motor functions.

## Methods

**Experimental subjects**. We recorded data from three 9–10 kg male *Macaca mulatta* monkeys (J, C, T; age: 6, 14, and 11 years when the experiments started) while they performed two sets of two tasks of wrist or reach-to-grasp motor tasks over several sessions (see Tasks, below). All surgical and behavioral procedures were approved by the Animal Care and Use Committee at Northwestern University. The monkeys were implanted with a 96-channel microelectrode silicon array (Utah electrode arrays, Blackrock Microsystems, Salt Lake City, UT) in the hand area of M1, which we identified intraoperatively through microstimulation of the cortical surface. For monkey C, we recorded neural activity for each of the two sets of tasks using different microelectrode arrays sequentially implanted in different brain hemispheres. The monkeys were also implanted with intramuscular EMG electrodes in a variety of wrist and hand muscles. We report data from the following muscles: Monkey J: flexor carpi radialis (FCR), flexor carpi ulnaris (FCU), extensor carpi radialis (ECR), extensor carpi ulnaris (ECU), flexor digitorum profundus (FDP), flexor digitorum superficialis (FDS), extensor digitorum communis (EDC; radial and ulnar aspects), brachioradialis, and supinator; Monkey C: FCR, FCU, ECR, ECU, FDP (radial and ulnar aspects), FDS (radial and ulnar aspects), EDC (radial and ulnar aspects), flexor pollicis brevis (FPB), opponens pollicis, and extensor pollicis longus; Monkey T: ECR, ECU, FCR, FCU, FDP (radial and ulnar aspects), FDS (radial and ulnar aspects), EDC, FPB, first dorsal interosseous (FDI). For the wrist tasks of monkey C, we recorded EMGs using pairs of gelled surface electrodes placed over FCR, FCU, ECR, ECU, FDS, and EDC. We report EMG results for 5–12 muscles (mean ± s.d., 8.2 ± 2.8), depending on the session. All surgeries were performed under isoflurane gas anesthesia (1–2%) except during cortical stimulation, for which the monkeys were transitioned to reduced isoflurane

(0.25%) in combination with remifentanil (0.4 μg kg$^{-1}$ min$^{-1}$ continuous infusion). Monkeys were administered antibiotics, anti-inflammatories, and buprenorphine for several days after surgery.

**Tasks and recordings**. In each session, monkeys performed either a set of reach-to-grasp tasks or a set of wrist tasks (Fig. 2). Within a session, the tasks were performed in consecutive blocks; the order of the tasks varied randomly across sessions. For each set of tasks, we collected data from two monkeys during two or three sessions. This shows reproducibility in the standard manner for this type of research; no blinding was necessary. All monkeys had been trained prior to their implant surgeries, and were proficient at the tasks at the time of the recordings. Monkeys C and T performed the set of reach-to-grasp tasks, which comprised the "ball" task and a power grip ("grip") task (monkey C, three sessions; monkey T, two sessions). In the ball task, monkeys had to reach to a ball (diameter 24, 35, or 40 mm), grasp it, and then transport it and drop it in an open cylindrical container[55]. In the power grip task, monkeys reached to and grasped a pneumatic tube that then had to be squeezed to control the movement of a cursor used to acquire one of two or three 1-D force targets[55]. Monkeys initiated both tasks by resting their hand on a touch pad, and waited for a target—or go signal for the ball task—to be presented. Monkeys C and J performed the wrist tasks, which comprised three 1-D tasks[2,4,53]: an isometric task, a movement task, and an elastic loaded movement task (both monkeys, three sessions); monkey J also performed a 2-D isometric center-out task[12] in two of three sessions (see Fig. 2a, b). Throughout the paper, we abbreviate these tasks as "iso," "mov," "spr," and "iso2D," respectively. As for the reach-to-grasp tasks, monkeys could initiate movement after the target was presented. Note that we recorded the wrist and reach-to-grasp datasets from monkey C in different years, using microelectrode arrays implanted in different cortical hemispheres. During the experiments, we recorded neural and EMG data, as well as kinematics or force, depending on the task. All data were saved to disk and analyzed in Matlab (The Mathworks Inc., Natick, MA).

To characterize neural population activity, we identified threshold crossings from each electrode; these included well-discriminated single-unit as well as multi-unit activity. Throughout this paper we refer to these as neural units, without distinction. For each session, data included all units whose average waveform, triggered by the threshold crossing, remained stable across all tasks (examples in Fig. 2, Supplementary Fig. 2). We did not choose neurons based on tuning, modulation depth, or any other property. To obtain a smooth discharge rate as function of time, we applied a Gaussian kernel (s.d.: 50 ms) to the binned square-root-transformed firings (bin size: 20 ms) of each unit[34]. For each task, this produced a neural data matrix $X$ of dimensions $n$ by $T$, where $n$ is the number of recorded units and $T$ is time duration of all concatenated trials. It must be noted that although our analyses were based on threshold crossings, a recent simulation and experimental study[57] indicates that the properties of the manifold and the activity within it would not have been different if we had used well-isolated single units instead.

The EMG envelopes, a proxy for the neural commands to muscles, were computed by a sequence of high-pass filtering (fourth-order zero-phase Butterworth filter, $f_c$: 10 Hz), rectification, and low-pass filtering (fourth-order zero-phase Butterworth filter, $f_c$: 50 Hz) of the raw EMG signals. We subsequently normalized these EMG envelopes by the 99th percentile of their distribution across all tasks for each session.

We used single-trial data for all the analyses except for dPCA, a method that requires trial-averaged data[51] (see details below). A trial was defined from target presentation (or go signal for the ball task) until the monkey received a reward; the very few unsuccessful trials were discarded. For trial averaging, we computed the mean firing rate (peristimulus time histogram) from target presentation until an end time determined by the shortest time to reward. We used both the reach-to-grasp and wrist datasets for all the analyses except for dPCA; the latter requires target equalization across tasks, which can only be achieved for the wrist tasks (see main text and dPCA section below). In every session, we made all possible comparisons between pairs of tasks.

**Task-specific neural manifolds and latent activity**. The activity of $n$ recorded units was represented in a neural space, an $n$-dimensional sampling of the state of M1. In this space, each point represents the state of the population of recorded units, and the coordinate of this point along each axis represents the firing rate of the corresponding unit (Fig. 1b). Within this space, we computed the low-dimensional neural manifold associated with each task by applying PCA to the corresponding smoothed firing rates of all $n$ units. In an $n$-dimensional space, PCA finds $n$ principal components (PCs), each a linear combination of the firing rates of the recorded units, designed to sequentially maximize the amount of shared variance (covariance). The PCs are ranked according to their contribution to explaining the total amount of variance in the original data. We defined $m$-dimensional task-specific manifolds that accounted for most of the neural population variance by keeping only the leading $m$ PCs (Fig. 1b). We chose $m = 12$, to account for at least 60% of the total neural variance across all tasks and monkeys (Supplementary Fig. 3). Importantly, and in agreement with previous reports[29,38,58], the results reported here were not sensitive to the somewhat arbitrary choice of manifold dimensionality (see Supplementary Fig. 5a for results for $m = 8, 15$). Each PC is a neural mode associated with a specific direction within the

neural space; together, the neural modes provide a basis that spans the low-dimensional neural manifold. We computed the latent activity, the dynamic activation of the neural modes, by projecting the $n$-dimensional, time-varying neural population activity onto each of the $m$ neural modes (PCs) that span the neural manifold.

**Comparison of task-specific neural manifolds.** Principal angles provide a measure of the relative alignment of two $m$-dimensional manifolds embedded in an $n$-dimensional space; their alignment is quantified in terms of the $m$ angles between sequentially aligned pairs of basis vectors, each within one of the respective manifolds[48]. These vectors, selected in each manifold so as to systematically minimize the angle between them, provide a new basis for each of the two manifolds being compared. Note that manifold directions chosen to minimize the angles between manifolds are not necessarily those that maximize variance within each of the two manifolds; it is thus not the angles between the PC neural modes that determine the principal angles. Our hypothesis that the task-specific manifolds compared here are similarly oriented implies that the leading principal angles will be small.

To compute the principal angles between two $m$-dimensional manifolds $A$ and $B$ embedded in an $n$-dimensional neural space, we follow the method by Björck and Golub[48]: consider the corresponding $m$-dimensional bases $\mathbf{W}_A$ and $\mathbf{W}_B$ provided by the $m$ leading PC neural modes, construct their $m$ by $m$ inner product matrix, and perform a singular value decomposition to obtain

$$\mathbf{W}_A^T \mathbf{W}_B = \mathbf{P}_A \mathbf{C} \mathbf{P}_B^T \qquad (1)$$

Here $\mathbf{W}_i$, $i = A, B$ are the $n$ by $m$ PC matrices that span the task-specific manifolds $A$ and $B$; the corresponding PC neural modes are their column vectors. The matrices $\mathbf{P}_A$ and $\mathbf{P}_B$, both of dimension $m$ by $m$, define the new manifold directions that successively minimize the principal angles. Note that these new projections are specific to the pair of tasks being compared. The matrix $\mathbf{C}$ is a diagonal matrix whose elements are the ranked cosines of the principal angles $\theta_i$, $i = 1,\dots, m$:

$$\mathbf{C} = \mathrm{diag}(\cos(\theta_1), \cos(\theta_2), \dots \cos(\theta_m)) \qquad (2)$$

Note that, by construction, the principal angles are ordered form smallest to largest.

To assess whether the experimentally obtained principal angles between pairs of task-specific manifolds were significantly small, we compared them to the principal angles between manifolds associated to surrogate datasets that were randomly generated but preserved key statistics of the actual data. These surrogate datasets were generated using the Tensor Maximum Entropy (TME) method[27], to preserve the structure of the covariance of the original neural data both over time and across targets. We generated and compared 10,000 surrogate pairs for each pair of tasks. For each pair of surrogate datasets, we computed their associated neural manifolds and the principal angles between them, using the same methods as for the original neural data. We used the 0.1th percentile of the distributions of principal angles between surrogate datasets to define a threshold below which angles can be considered significantly small (with a probability $P < 0.001$). We note that even though the TME method is designed for trial-averaged data[27], we were able to apply it to assess the significance of the neural manifolds identified using concatenated single-trial data because the orientation of each task-specific neural manifold showed only small differences when computed using either type of data.

An additional analysis to compare the orientation of manifolds from two different tasks was based on measuring how much of the neural variance associated with a given task was accounted for when the data was projected onto the manifold of a different task[58]. We computed the variance accounted for (VAF) when projecting the data from task $A$ onto the manifold of task $B$ (across-task VAF) and compared it to the VAF obtained when the data from task $A$ were projected onto the original manifold of task $A$ (within-task VAF). If the two manifolds $A$ and $B$ had similar orientations within the high-dimensional neural space, we would expect these across-task to within-task VAF ratios to be quite close to one. To verify that the computed VAF ratios were significantly large, we compared them to the largest ratios that could be obtained by chance. For each task comparison, we projected the original data for task $A$ onto 10,000 randomly generated and thus randomly oriented $m$-dimensional manifolds, obtaining a distribution of surrogate across-task VAFs. We used the 0.1th percentile of this distribution as the maximum across-task VAF that could be expected by chance, and computed the corresponding ratio to the within-task VAF. We assessed the significance of the actual across-task to within-task VAF ratios by comparing them to these surrogate VAF ratios. We used a two-sided Wilcoxon rank sum test, because the data did not conform to normality (Lilliefors test, $P < 0.01$ to reject). Note that the comparison was performed twice for each pair of tasks, considering either $A$ or $B$ as the within-task manifold.

**Task-specific and task-independent latent activity.** To understand the role of the latent activity in movement generation, we used another linear dimensionality reduction method, dPCA[51]. This approach identifies a single neural manifold for all the data (here, pooled across all tasks in one session), spanned by neural modes

whose activity is a linear readout of the activities associated with specifically chosen behavioral parameters[51]. The ability to find a single neural manifold for all the tasks in one session is due to the strong similarity in the orientation of the corresponding task-specific manifolds (Fig. 3).

Mathematically, dPCA starts by representing the mean-subtracted, trial-averaged activity of all units, concatenated over all tasks and targets within a session, as a neural data matrix $\bar{X}$ of dimensions $n$ by $(\alpha \times \gamma \times \tau)$. Here $n$ is, as before, the number of recorded units, $\alpha$ is the number of tasks performed in that session, $\gamma$ is the number of targets equalized across tasks, and $\tau$ is the trial time duration, equalized across tasks and targets by truncation to the shortest trial. Since trial durations were overall very similar, the deleted portions were short. This data matrix $\bar{X}$ is decomposed as a sum of matrices $\bar{X}_\emptyset$, each describing the neural activity associated with a specific behavioral parameter $\emptyset$, and the measurement noise $\mathbf{X}_{\mathrm{noise}}$:

$$\bar{\mathbf{X}} = \sum_\emptyset \bar{\mathbf{X}}_\emptyset + \mathbf{X}_{\mathrm{noise}} \qquad (3)$$

The decomposition ensures that the $\bar{X}_\emptyset$ are uncorrelated: the $n$ by $n$ covariance matrix $\mathbf{C} = \bar{\mathbf{X}}\bar{\mathbf{X}}^T$ is thus the sum of covariance matrices, one associated with each behavioral parameter:

$$\mathbf{C} = \sum_\emptyset \mathbf{C}_\emptyset + \mathbf{C}_{\mathrm{noise}} \qquad (4)$$

Dimensionality reduction in dPCA is based on the minimization of a reconstruction error

$$\mathbf{E}_{\bar{\mathbf{X}}} = \sum_\emptyset \mathbf{E}_{\bar{\mathbf{X}}_\emptyset} \qquad (5)$$

with

$$\mathbf{E}_{\bar{\mathbf{X}}_\emptyset} = \left\| \bar{\mathbf{X}}_\emptyset - \mathbf{A}_\emptyset \bar{\mathbf{X}} \right\|^2 \qquad (6)$$

The minimization of the reconstruction error becomes equivalent to a classical regression problem with a least-squares solution:[51]

$$\mathbf{A}_\emptyset^{\mathrm{LS}} = \bar{\mathbf{X}}_\emptyset \bar{\mathbf{X}}^T (\bar{\mathbf{X}}\bar{\mathbf{X}}^T)^{-1} \qquad (7)$$

In dPCA, the rank $m$ of the $n$ by $n$ matrix $\mathbf{A}$ is chosen as the desired dimensionality of the manifold. The least-square problem thus becomes a reduced-rank regression problem that is solved using singular value decomposition[51]. A detailed description of dPCA for neural population data has been given by Kobak et al.[51]; notably, this implementation of dPCA has an analytic as opposed to a numerical solution.

The behavioral parameters $\emptyset$ used here are: time along the trial, task, target location, and the combination task/target location. We performed the dPCA analysis on the wrist tasks, as these datasets included three or four tasks ($\alpha = 3$ or $\alpha = 4$) for which six targets ($\gamma = 6$) were similarly located in space (see target organization in Fig. 4d, Supplementary Fig. 1 and 2), thus easily achieving target equalization. As before, the chosen manifold dimensionality was $m = 12$, although the results held for $m = 8, 15$ (see Supplementary Fig. 5b). In spite of the constraint that the activity of each dPC has to covary with one or a few of the chosen behavioral parameters, the neural variance gradually explained by the neural modes identified with dPCA was very similar to that explained by the PCA modes (see Supplementary Fig. 6b).

**Comparison of latent activity across tasks.** To investigate potential similarities in latent activity across tasks, we compared the corresponding task-specific manifold activity using CCA[52]. The method systematically finds new directions within each manifold such that the corresponding one-dimensional projected activities are maximally correlated. As is the case with the manifold directions used to compute principal angles, these directions are not necessarily those of the PC neural modes selected to maximize projected variance.

Consider again the two manifolds $A$, $B$ to be compared. We start by projecting the neural activity for each task onto the $m$ PC neural modes that span the respective manifolds, to obtain two $m$ by $T$ latent activity matrices $\mathbf{L}_A$ and $\mathbf{L}_B$; here $T$ is time duration of all concatenated trials for each task within a session. CCA finds two linear transformation matrices, one for each of the two matrices $\mathbf{L}_i$, $i = A, B$, to obtain new manifold directions so that the activities projected onto these new directions are maximally correlated across the two manifolds[52].

CCA starts with a QR decomposition of the transposed latent activity matrices $\mathbf{L}_A$ and $\mathbf{L}_B$, $\mathbf{L}_A^T = \mathbf{Q}_A \mathbf{R}_A$, $\mathbf{L}_B^T = \mathbf{Q}_B \mathbf{R}_B$. The first $m$ column vectors of $\mathbf{Q}_i$, $i = A, B$ provide an orthonormal basis for the column vectors of $\mathbf{L}_i^T$, $i = A, B$. We then construct the $m$ by $m$ inner product matrix of $\mathbf{Q}_A$ and $\mathbf{Q}_B$ and perform a singular

value decomposition to obtain

$$\mathbf{Q}_A^T \mathbf{Q}_B = \mathbf{U}\mathbf{S}\mathbf{V}^T \qquad (8)$$

The elements of the diagonal matrix $\mathbf{S}$ are the canonical correlations (CCs), sorted from largest to smallest. The new manifold directions that CCA finds so as to maximize the pairwise correlations between latent activities across the two tasks are the corresponding $m$ by $m$ matrices

$$\mathbf{M}_A = \mathbf{R}_A^{-1}\mathbf{U}, \mathbf{M}_B = \mathbf{R}_B^{-1}\mathbf{V} \qquad (9)$$

To implement this method, the matrices $\mathbf{L}_i$, $i = A, B$ included all the concatenated trials for each of the two tasks being compared. To assemble these data matrices, we first equalized the number of trials across all tasks and targets within the corresponding session; we took the first $k$ trials for each task and target, with $k$ being the minimum number of successful trials across all targets and tasks within the session. When comparing tasks with different number of targets, the number of trials per target was adjusted so as to equalize the number of trials per task. For each trial, we used either a 700 ms long (wrist tasks) or a 1000 ms long (reach-to-grasp tasks) window of neural data, starting around target onset or go signal. We then concatenated these $k$ individual trials according to the common sequence of visual targets. When comparing two 1-D wrists tasks, we matched the trials by target location; when comparing the 2-D isometric task to any of the 1-D wrist tasks, we labeled targets according to their projection onto the horizontal axis. No target matching was done for the reach-to-grasp tasks, as the ball task had no targets. We did not exclude trials based on their execution time, or based on the EMG, kinematics, or force patterns; only the few failed trials were excluded.

We used an analysis of within-task variability across trials to obtain an upper bound for the across-task CCs. We computed within-task CCs by dividing all the trials for a given task into two random target-matched subgroups (100 repetitions) across which we calculated the corresponding CCs. We used the 99.9th percentile of each within-task CC distribution as the upper bound CC value; for a given pair of tasks, the across-task upper bound was taken to be the maximum of the two corresponding within-task upper bounds. To more clearly visualize how the across-task CCs relate to this upper bound, we computed the ratio between them (Supplementary Fig. 7e). We also used bootstrapping to assess the significance of the across-task comparisons of latent activities. The latent activity for one of the tasks was randomized over time (for all modes simultaneously), and smoothed with a Gaussian kernel designed to match the spectral content of the actual data (s.d.: 50 ms). We then used CCA to find maximal correlations between these surrogate first task data and the data from the second task. We repeated this procedure 5000 times for each task comparison, and took the 99.9th percentile of the resultant distribution of CCs as a significance threshold ($P < 0.001$).

**Neural variance explained**. We used a series of analyses (principal angles, CCA, dPCA) to compare the structure of the neural modes and their activity across different tasks. Each one of these approaches involves projecting the original $n$-dimensional data onto manifold directions that differ from those found by PCA. In order to assess the cumulative neural variance associated with incremental subsets of these new directions $1 \leq h \leq m$, we express the fractional variance accounted for (VAF) in terms of the corresponding reconstruction error,

$$\text{VAF}_h = \frac{\|\mathbf{X}\|^2 - \|\mathbf{X} - \mathbf{D}_h\mathbf{E}_h\mathbf{X}\|^2}{\|\mathbf{X}\|^2} \qquad (10)$$

For principal angles and CCA, the $n$ by $T$ data matrix $\mathbf{X}$ is task-specific and represents concatenated trials for a given task during one session, pooled across all targets. The PCA leading to the $m$-dimensional manifold requires that the data be mean-subtracted, $\Sigma_j x_{ij} = 0$ for all $i$. Thus $\|\mathbf{X}\|^2 = \sum_{ij} x_{ij}^2$ is the total variance. The matrices $\mathbf{E}_h$ and $\mathbf{D}_h$ are encoding and decoding matrices, respectively. The $h$ by $n$ matrix $\mathbf{E}_h$ projects the original data onto the leading $h$ new manifold directions, and the $n$ by $h$ matrix $\mathbf{D}_h$ optimally reconstructs the data from this $h$-dimensional projection. For both principal angles and CCA, encoding starts by projecting the original data onto the manifold spanned by the $m$ leading PCs: $\mathbf{W}_i^T\mathbf{X}_i$ for $i = A, B$.

For principal angles, the data is then projected into new manifold directions given by the columns of the corresponding matrix $\mathbf{P}_i$, $i = A, B$. The encoding matrix for this specific $A,B$ comparison thus is $\mathbf{E}_h = \mathbf{P}_h^T\mathbf{W}^T$, where $\mathbf{P}_h^T$ is the transpose of the first $h$ columns of the projection matrix $\mathbf{P}$. Because both the matrix $\mathbf{W}$ and the matrix $\mathbf{P}$ are orthogonal, the optimal decoding matrix—in the sense of minimizing the squared reconstruction error—is the transpose of the encoding matrix. Thus $\mathbf{D}_h = \left(\mathbf{P}_h^T\mathbf{W}^T\right)^T = \mathbf{W}\mathbf{P}_h$. To represent the variance explained across the two tasks $A$ and $B$ as a single number, we took the mean of the two values.

For CCA, once the data has been projected onto the $m$ leading principal components using the corresponding task-specific PC matrix $\mathbf{W}_i$, $i = A, B$, it is then projected into new manifold directions given by the columns of the corresponding matrix $\mathbf{M}_i$, $i = A, B$. The encoding matrix for this specific $A, B$ comparison thus is $\mathbf{E}_h = \mathbf{M}_h^T\mathbf{W}^T$, where $\mathbf{M}_h^T$ is the transpose of the first $h$ columns of the projection matrix $\mathbf{M}$. Because the matrix $\mathbf{W}$ is orthogonal but the matrix $\mathbf{M}$ is not, the optimal

decoding matrix is not just simply the transpose of the encoding matrix, but $\mathbf{D}_h = \mathbf{W}\mathbf{M}_h\left(\mathbf{M}_h^T\mathbf{M}_h\right)^{-1}$. The extra factor of $\left(\mathbf{M}_h^T\mathbf{M}_h\right)^{-1}$ corrects for the non-orthogonality of the $\mathbf{M}$ matrix.

The amount of explained variance for dPCA is similarly computed through the reconstruction error of the mean-subtracted trial-averaged data matrix $\bar{\mathbf{X}}$,

$$\text{VAF}_h = \frac{\|\bar{\mathbf{X}}\|^2 - \|\bar{\mathbf{X}} - \mathbf{D}_h\mathbf{E}_h\bar{\mathbf{X}}\|^2}{\|\bar{\mathbf{X}}\|^2} \qquad (11)$$

The encoding and decoding matrices $\mathbf{E}_h$ and $\mathbf{D}_h$ are computed as described by Kobak, Brendel and colleagues[51]. When computing the dPCA manifold for a specific task, the matrix $\bar{\mathbf{X}}$ included only the trial-averaged neural activity for the task being considered, concatenated over all corresponding targets. In contrast with principal angles and CCA, dimensionality reduction is directly effected by dPCA and does not require PCA as a preliminary step. In addition, the analysis is task specific and does not involve a comparison between two tasks. Note that since the manifold directions found by dPCA are not necessarily orthogonal, the decoding matrix $\mathbf{D}_h$ is not the transpose of the encoding matrix $\mathbf{E}_h$.

**Relationship between latent activity and EMGs**. To understand the role of the neural modes in movement generation, we investigated how their activity related to the ongoing muscle commands (EMGs) by building standard neural decoders as previously used by our group[61]. We were particularly interested in the role of the target-related but task-independent neural modes identified with dPCA. To assess whether these target-related modes captured task-independent EMG components, we compared the predictions of decoders based on target-related but task-independent modes as inputs to the predictions of decoders that used as inputs the activity of the other three sets of modes (time-related, task-related, and task-target-related).

The neural to EMG decoders were multiple-input single-output linear filters followed by a static non-linearity:

$$y(t) = \sum_{k=1}^{K} \sum_{\tau=0}^{M-1} h_k(\tau)L_k(t - \tau) \qquad (12)$$

$$z(t) = a + b \cdot y(t) + c \cdot y^2(t) \qquad (13)$$

where $z(t)$ is the predicted EMG, obtained by applying a static non-linearity to the output $y(t)$ of the linear model, which estimated the EMG as a linear combination of the current and past values of the latent activities, $L_k$. In this case two inputs, $k=1, 2$, were weighted by coefficients $h_k(\tau)$, where $\tau$ represents a time lag into the past. The coefficients $h_k(\tau)$ followed from the lagged auto-correlations and cross-correlations of the decoder inputs and outputs[61]. The coefficients $a$, $b$, and $c$ of the second-order polynomial in the static non-linearity resulted from least-squares error minimization.

We built a single decoder for each behavioral parameter $\emptyset$ using data from all the tasks the monkeys performed during one session. We assessed the quality of fit on single trial data in terms of the normalized coefficient of determination, which is the ratio of the $R^2$ of the EMG predictions based only on the neural modes related to the specific behavioral parameter $\emptyset$, to the $R^2$ of the EMG predictions based on all 12 neural modes. Fits were cross-validated (leave-one-out multifold cross-validation using 30 s folds) in all cases. We compared EMG predictions for different behavioral parameters $\emptyset$ using a paired two-sided Wilcoxon rank sum test including each fold.

To interpret our decoding results, we decomposed the EMGs from all the tasks within a session into EMG modes using dPCA. We kept $m = 4$ modes, obtained using the same method as for the dPCA of neural data. Four dPCs sufficed since across all datasets, $3.2 \pm 0.8$ EMG modes (mean ± s.d.) explained ≥95% of the total EMG variance for each wrist task separately (Supplementary Fig. 1d). When predicting subsets of EMG modes, we used decoders with the same structure as described above, and followed the same cross-validation procedure.

**Computational model**. Given that M1 has extensive projections to upper limb muscles, and that M1 spiking is strongly predictive of EMGs, we asked whether the observed similarities in latent neural activity could be readily explained by similarities in the EMGs. To analyze this possibility, we constructed a model that assumes a fixed relationship between M1 activity and muscle activities. We used actual EMG data for a given pair of tasks to obtain the corresponding simulated neural activity. We then used the same methods as described for the actual neural data to obtain the task-specific neural manifolds and their associated latent activity, and to compare these across tasks for the simulated neural data. Finally, these across-task similarities in latent activities for the simulated data were compared to those obtained for the actual data.

Our model, largely based on that in ref. [70], is as follows. Each neural unit $j$ was simulated as a Poisson process with a time-dependent mean $\lambda_j(t)$ given by

$$\lambda_j(t) = a_j + \sum_i b_{ji}\varepsilon_i(t) + \eta_j(t) \qquad (14)$$

with $0 \le a_j \le 0.1$ and $-1 \le b_{ji} \le 1$. Here, $\varepsilon_i(t)$ is the time-dependent EMG of the $i$th recorded muscle, and $\eta_j(t)$ is additive Gaussian noise with zero mean and a variance of 0.05. For each task, we used all the recorded EMGs to simulate as many units as we had recorded experimentally. The $\{\lambda_j\}$ were scaled so as to match the mean firing rates of the actual neural data. In addition, we checked that second-order statistics of the simulated firing as measured by the VAF followed a similar trend to that of the real data (Fig. 5e) across a fairly broad range of model parameters $\{k_j, g_{ij}\}$. Finally, we used CCA to compare the activities of these simulated neural modes across every pair of wrist tasks that monkeys performed in each session (Fig. 5f).

**Control analyses.** To probe the dependence of manifold geometry and activity on the dimensionality of the embedding neural space, we performed unit-dropping numerical experiments. To check for orientation robustness, we selected from all recorded units two random subsets with the same number of units each. We created two sets of population activity; in each of them, the activity of the units chosen for dropping was correspondingly set to zero. In each set, we used PCA to obtain the 12-D manifolds spanned by the 12 leading PCs; both manifolds were embedded within the original neural space. We computed the principal angles between them; small principal angles would signal orientation robustness against the particular choice of units. We repeated this procedure dropping 10%, 20%, 30%, and 40% of all recorded units (100 random pairs in each case), to also check for robustness against the number of recorded units (Supplementary Fig. 3e). In all cases, the dimensionality of the data was preserved as that of the original neural space, thus allowing us to compare manifold orientations using principal angles.

To check for latent activity robustness, we again selected a random subset of all recorded units, set their activities to zero, performed PCA on the resulting neural ensemble, obtained the 12-D manifold spanned by the 12 leading PCs, and projected the population activity onto these neural modes to obtain their activity. We then used CCA to compare the activities of these modes to the activities of the 12 leading modes computed from all recorded units. We repeated this operation dropping 10, 20, 30, and 40% of all recorded units (100 random pairs in each case). We expected leading CCs close to 1, indicating latent activities that were robust against specific choice of units and, to some extent, their number (Supplementary Fig. 3f).

We also evaluated whether our results depended on the choice of manifold dimensionality, by replicating the main analyses for neural manifold dimensionalities $m = 8$ and 15. No qualitative changes in the results were observed as $m$ was changed (Supplementary Fig. 5).

To assess EMG stability across tasks, we computed the principal angles between the EMG manifolds for two different tasks, to establish whether they were as well aligned as the corresponding neural manifolds. To quantify the across-task stability of latent activity we applied CCA to the EMG activity associated with different tasks, and computed the ratio of the across-task CCs in latent neural activities to the across-task CCs in EMGs for as many dimensions as EMGs we had available (typically less than 12 well-recorded muscles in any given session). Task-specific EMG manifolds, pairwise principal angles between them, and across-task comparisons for latent muscle activities were computed using the same methods as for neural activity.

**Code availability.** Code packages for comparing manifold orientation, comparing latent activity, simulating neural activity, and EMG decoding are available upon request to the authors. For dPCA, we used the publicly available toolbox from the Machens lab[51] (https://github.com/machenslab/dPCA). For TME, we used the publicly available toolbox from the Cunningham lab[27] (https://github.com/gamaleldin/TME).

## Data availability

All relevant data are available from the authors upon request.

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

## Acknowledgements

This work was supported in part by Grant FP7-PEOPLE-2013-IOF-627384 from the Commission of the European Union (J.A.G.), by Grant F31-NS092356 from the National Institute of Neurological Disorder and Stroke and Grant T32-HD07418 from the National Center for Medical Rehabilitation Research (M.G.P.), by Grant DGE-1324585 from the National Science Foundation (S.N.N.), by Grant 22343 from the Fonds de Recherche du Québec–Santé (C.E.), and by Grant NS053603 from the National Institute of Neurological Disorder and Stroke (S.A.S. and L.E.M.).

## Author contributions

J.A.G., S.A.S., and L.E.M. devised the project. S.N.N. and C.E. performed the experiments. J.A.G. analyzed the data. J.A.G., M.G.P., S.A.S., and L.E.M. interpreted data and wrote the manuscript.
