## [Peer Review File · Nature Communications]

Reviewers' comments:

Reviewer #1 (Remarks to the Author):

This is a potentially interesting paper that attempts to compare neural manifolds across a range of tasks. Overall, it is an important area of investigation. As outlined below, however, there are several issues that limit enthusiasm. The primary concern is that multiple tasks were used and multiple related analytical approaches were employed without an apparently clear unifying framework. There is mention that the different methods may be similar, it would help to better explain the fundamental approach of using multiple yet related methods. This is particularly challenging as multiple tasks are used in only a subset of animals. Ideally, the same set of tasks would have been used; it would then be easier to compare single neural responses and the population dynamics. It would also better motivate exactly what the population dynamics represent relative to individual neurons. While this is not the case here, it would help to frame the manuscript from that perspective.

1. It is important to show that the single units (criteria used?) show similar results to the MU data. For example, it is quite possible that MUs appear to show common modes only because they are MU. If only SUs are used, is there common variance?
2. Better justification for 12 dimensions is important. It is surprising that only 60% of the variance is accounted for. Would this result hold for higher variance accounted for?
3. What does PCA of the EMG show? What is the variance accounted for relative to number of components?
4. Figure 1 is confusing given that multiple tasks are used. I would recommend making a more complete Figure 1 that can account for all the tasks used. It would help to also show the types of actions and how they might relate (e.g. isometric only vs wrist vs wrist/grasp).
5. Figure 1g. This could be the result of lumping in multiple conditions. What about comparing more similar vs more different conditions?
6. Do the result hold if comparing 'more similar' vs 'more different' actions. In other words, comparison of two wrist based tasks vs a wrist and wrist/finger task. Again, this might help identify how the subspaces relate to the nature of task itself. As mentioned in the manuscript, comparisons of rodent walking and reaching show evidence of orthogonality.
7. When using PCA, what were the distribution of weights in the PCs? Are the units with the highest weights similar across task (i.e. correlation of activity, this is related to point #5)?
8. The analysis of angles between modes requires additional work. The manuscript uses random data to determine "small values". However, I could imagine that a more realistic dataset could result in a different conclusion. Could sufficiently scrambled neural data be used? The recent Cunningham lab (Nature Neuroscience) work could offer a strategy where only higher order statistics are perturbed (e.g. preserving the pairwise correlation matrix).
9. For CCA analysis, what is the VAF for the first 3 modes?

Minor.

1. The manuscript used fairly dramatic language. For example, the word critical is used often. It is not clear, at least to me, that all of these are warranted. Consider reducing bombast...
2. Page 22: Reference 50/43 appear to be incorrect. These refer to methods used in this manuscript.
3. Supp Figure 2. Scale bars are not shown. What about raw waveforms (i.e. not only the mean). ISI distribution?
4. Page 24. How was the firing rate smoothed?

Reviewer #2 (Remarks to the Author):

Review

Multiple tasks viewed from the neural manifold: Stable control of varied behavior

This paper hypothesizes the control of a variety of movements as arising from population neural activity in the primary motor cortex (M1) that can be well explained by a reduced set of neural 'modes'. The idea of a reduced-dimension neural manifold to represent the neural activity has been shown to exist in different settings, with an insightful review [1] by a subset of the authors containing some of the key references. The key question being addressed in this paper is : how closely aligned are the neural manifolds in a variety of different tasks, i.e. how far away from orthogonal are they? Indeed, if the neural manifolds perfectly overlap during a wide variety of tasks, and the only difference between the tasks is the dynamics of the activity within these manifolds, it would be an extremely insightful finding and may lead to some important discoveries about the structure of connections in M1. The analyses performed in this paper seem to suggest that the manifolds are indeed aligned at a significant level across a variety of tasks. The second question being addressed in this paper is : does the neural activity evolve similarly or differently in these lower dimensional neural manifolds, across tasks? The authors use two analysis techniques to answer this question. The first analysis method (CCA) leads to the result that the neural activity evolves in a fairly similar manner across tasks, which is subsequently clarified by a second analysis method (dPCA) that finds a substantial component of the activity to evolve in a task-independent manner, while the rest to evolve in a task-dependent manner. Lastly, this paper addresses the question: can the muscle activity be explained or decoded by the lower dimensional neural manifold? Indeed, this is shown to be possible to some extent in these tasks. Overall, the main finding of Gallego et al. is the similarity of the neural manifolds during a variety of tasks, with the additional analyses serving to complement the main hypothesis and add context to the datasets.

The question of the similarity of the neural manifolds is indeed an interesting one. Although it has not been explicitly studied as such, we can infer from past studies that simple, classic motor tasks (for example, reaching in one of eight different directions) operate in aligned neural manifolds [2, amongst others]. Conversely, it has been shown in one study that performing two very different tasks leads to nearly orthogonal neural manifolds being recruited (contrasting results from this paper) [3]. Moreover, it has been shown that preparatory activity in the M1 lies in an orthogonal subspace from the M1 activity during movements [4]. At which task complexity and / or task diversity the neural manifolds move from aligned to orthogonal in the M1 is as yet unknown. As such, if additional analyses are performed to strengthen the main result in this paper showing that a wide variety of tasks operate in aligned neural manifolds, it would be a very exciting result and help to elucidate the role and organization of M1 neurons in performing skilled movements.

This paper analyzes data collected from 3 different monkeys performing 6 different tasks; a subset of the tasks were performed by each monkey. While the results would naturally be more convincing with a larger variety in tasks, the tasks used in this paper were sufficiently different from each other to warrant this analysis.

Three main techniques were used to analyze the neural data – principal angles (PA), canonical correlation analysis (CCA), and demixed Principal Component Analysis (dPCA), all to capture the similarities and / or differences in a lower dimensional space of neural activity. Moreover, a decoder was built to illuminate the relationship between the neural activity in the manifold and the EMG activity. Although this seems a priori a thorough analysis of the datasets considered in this paper, below are some points that may help in making the results robustly supported at a level appropriate for this journal.

Major Concerns: In my view, this paper cannot be published until the following concerns are addressed.

- 1) Since the crux of the paper lies in the presence of small principal angles between the neural manifolds of different tasks, this analysis requires extra care. As the authors note, the threshold angles between 12D manifolds increased with the dimensionality of the neural space (Suppl. Fig. 5b). However, the fact that the thresholds increase considerably with n the number of neurons recorded is fundamentally problematic. This increase can be conceptualized as the following: the probability that two 2D vectors randomly sampled from a 3D space are orthogonal is lower than the probability that two 2D vectors randomly sampled from a 4D space are orthogonal. An alternative suggestion is to rely on permutations to compute the significance thresholds, as in [3], or, better yet, to draw random subspaces biased to the data covariance structure, as in [4] (see Supplementary Note 3 in [4]). This analysis will make the results of this paper much stronger. It is also key to have a discussion around the dependence of this threshold and the subsequent results on m , the number of PCs chosen, here 12. One suggestion is to show the analysis for varying m in a reasonable range, say 5 to 15.
- 2) An important question being asked here is the similarity of neural manifolds between a variety of tasks. That being said, the authors only compute the principal angles and perform the correlation analysis on pairs of 'similar' tasks. It would benefit the hypothesis if the principal angles between the three dimensional ("ball" and "grip" tasks) and the one dimensional tasks ("iso", "mov", "spr" tasks) are also small. It would also be interesting if these principal angles are larger than the principal angles between just the one dimensional tasks. This analysis may only be possible for one monkey (monkey C), but would go a long way towards showing that a larger variety of tasks have aligned neural manifolds. Without this, the reader is left wondering why this analysis was not performed.
- 3) It is key to have a thorough discussion of the variety in the different tasks in the context of past studies. For example, how different are the kinematics across tasks in this study, as compared to other motor studies mentioned in the references? The larger the variety in kinematics arising from the same or aligned neural manifolds, the more surprising and thus impactful are the results. The authors have shown the large variety in Suppl Figures 1 and 2, but a discussion of this variety in the context of past experimental studies is lacking. It is a strength of this paper to have a large variety of different tasks – this point can be stressed further throughout this paper.

Moderate Concerns: The following points are very strongly suggested before publishing this paper.

- 1) The three dimensional tasks, i.e. "ball" and "grip" are significantly more complex than the one and two dimensional tasks. A discussion on the difference in the activation dynamics in the neural manifold or differences in the nature of the neural manifold itself may be insightful. It would be beneficial to add an analysis on the amount of variance captured by the PCs calculated on the one dimensional task to the neural space occupied by the three dimensional tasks.
- 4) The analysis regarding the correlations in the 'neural mode dynamics' between different tasks, while interesting, paints an unfinished picture, which is subsequently clarified by the dPCA analysis. The across-task CCs as reported by the authors are not extremely high (though comparable to within-task CCs), as expected since a part of the neural activity does actually differ across tasks (otherwise task discrimination based on neural activity would be impossible). This point is clarified by the dPCA analysis, which shows that a significant part of the neural activity can be explained by task-independent effects, whereas the rest from task-dependent effects, while quantifying these effects. The CC analysis presumably captured the task-independent effects. However, it is unclear what additional information the CC analysis presents (a better motivation of the CC analysis might help clarify this point). It is clear that the dPCA analysis could not be carried out for a subset of the tasks, and thus a CC analysis would be warranted for these – though it is a suggestion that this may be moved to supplementary material.
- 5) As mentioned in the discussion (page 18) and corroborated by Reference 60 in the text, it is possible that the results in the paper are expected if I believe that the task-independent signal is the 'largest' signal in the neural activity. A discussion on this point is warranted earlier and much more prominently. Reference [5] may be relevant.
- 6) Although the dPCA analysis lends well to interpretability, more information could be provided on

how well the dPCs capture the per-task variance. Moreover, a discussion on whether the neural manifold identified by dPCA is close to those identified by the separate PCAs per task would be helpful.

7) Since the R^2 values for the target-related neural modes to the entire EMG activity in different tasks are modest, it is my opinion that the term "stable control of varied behavior" is perhaps an overstatement of the results. The target-related modes of the EMG can be decoded by the target-related neural modes fairly well, but the EMG activity contains other modes that the target-related neural manifold is apparently not entirely successful at decoding, which itself may be the cause of 'varied' behavior. Thus, the last sentence in the results section is accurate: the authors have "identified a stable, task-independent component in the mapping from specific neural modes to specific EMG modes". The title of the paper may not accurately reflect this result.

Minor Concerns: The following points are strongly suggested, but should not change the results or content of the paper too much.

- 1) The abstract should be modified to include a specific scientific question and the results of the paper in the context of other findings in the motor community.
- 2) Reference [5] may be relevant, along with the references 14 and 15 already listed in the paper in the second paragraph of the introduction (page 2).
- 3) It would be helpful to compare the results in the paper to existing results in the motor community in the last paragraph of the Introduction section.
- 4) In the CCA Methods section, the authors mention taking the average between the upper bounds of the CC distribution of two tasks (page 28). It is suggested to take the maximum instead.
- 5) The authors should detail how they equalized the number of trials across all tasks within the corresponding session, as mentioned on page 27 and page 30.
- 6) The bootstrapping method to assess significance in CCA analysis should be clarified and detailed (page 28).
- 7) It may be helpful to show or mention whether the dPCA results depend significantly on m .
- 8) The plots in Suppl Figure 3a warrant further discussion. For the wrist tasks, we see a relatively large number of neurons with high PC weights on the tail end of the neural modes, which is not seen in the reach-to-grasp tasks. It could be insightful to investigate why.
- 9) The authors use a specific static nonlinearity that seems arbitrarily chosen to decode the neural mode dynamics into the EMG activity. Did the authors try polynomials of different order, or different classes of nonlinearities? Motivating the second order polynomial is suggested. Moreover, the decoder is listed as linear in several other places in the paper, despite the presence of a nonlinearity.
- 10) Although the normalized R^2 values of the EMG predictions are listed, as calculated on page 31, the R^2 values of the EMG predictions based on all 12 neural mode dynamics are not listed, which would be helpful as a comparison.

Readability Concerns:

- 1) The authors seem to use 'activation dynamics', 'time varying activation', and 'neural mode dynamics' interchangeably. For the sake of readability, it is suggested that the authors stick to one of these terms, or clarify the differences between them.
- 2) Modify the first sentence of the second paragraph on page 3 to clarify that the hypothesis is that the same or aligned neural modes are activated during different tasks.
- 3) In Figure 1, is u_i the neural mode or the activation dynamics? The text would suggest neural mode. Also clarify this in the caption for Figure 1.
- 4) The second sentence of the first paragraph of the Discussion section reads awkwardly (page 16). "Critically, several of these studies... "
- 5) It is not certain that Reference 11 on page 18 is correctly placed.
- 6) There is a typo in first line of the third paragraph on page 27. hose → those.

References:

- [1] Gallego, J. A. et al. Neural Manifolds for the Control of Movement. *Neuron* (2017).
- [2] Churchland, M. M. et al. Neural population dynamics during reaching. *Nature* (2012).
- [3] Miri, A. et al. Behaviorally Selective Engagement of Short-Latency Effector Pathways by Motor Cortex. *Neuron* (2017).
- [4] Elsayed, G. F. et al. Reorganization between preparatory and movement population responses in motor cortex. *Nature Communications* (2016).
- [5] Elsayed, G. F. et al. Structure in neural population recordings: an expected byproduct of simpler phenomena? *Nature Neuroscience* (2017).

Reviewer #3 (Remarks to the Author):

In this work, the activity of tens of neurons is recorded in primary motor cortex (M1) while monkeys perform different manual tasks, including wrist isometric and movement tasks and grip tasks. The authors find that the neural modes identified from different tasks are similar, both in terms of the contribution of different neurons to the neural modes (PCA analysis) as well as the temporal dynamics (CCA analysis). Furthermore, they find task-independent neural modes (dPCA analysis) and relate them to muscle activity (EMG).

The data are extremely rich with the same neural data recorded during these different tasks. No previous study has compared M1 population activity recorded in as many different tasks. Furthermore, the authors take a modern view on M1 by characterizing the neural activity in terms of neural modes, rather than categorizing single neurons. For these reasons, this line of work has great potential. However, I have several substantial concerns (see details below).

Major comments:

1) It is unclear to what extent these results are simply a result of the EMG being similar for these different tasks. Let's say that there is a fixed relationship between M1 and muscles. Given the similarity of the EMG across tasks, would we expect to see the degree of similarity of the M1 activity across tasks that the authors report? In other words, have the authors found anything interesting in the neural activity beyond what can be deduced from EMG alone?

To address these questions, I have several concrete suggestions:

1a) If the analysis in Fig 3 is repeated on EMG, would we see a similar result?

1b) Supp Fig 7c is the only analysis which shows that there may be something interesting in the neural modes beyond what can be deduced from EMG. However, this result may be a trivial consequence of the fact that more neural modes (12) than EMG modes (7) are used in this analysis. Does the same result hold with an equal number of neural and EMG modes? Also, canonical correlations (like Pearson correlation) is sensitive to the amount of noise in the data. Is the amount of noise in the trial-averaged neural and EMG data similar?

1c) Construct a model in which M1 and EMG has a fixed relationship, and give the model the recorded EMG. Then show that the results found in this paper about M1 do not fall out of this model.

1d) Given that ~70% of the EMG variance is task-independent (Supp Fig 9), is it surprising that ~60% of the neural variance is task-independent (Fig 5c)? The model suggested above could help bolster

this claim.

2) One of the drawbacks of principal angles and canonical correlations is that they do not take into account the amount of variance in the data. In Fig 3a and 4b, the amount of variance in the data explained by the neural modes with low principal angles (Fig 3a) or high canonical correlations (Fig. 4b) should be indicated. If these neural modes explain large amounts of variance in the data, then the results can be interpreted as being more meaningful.

3) In Fig 3a, the chance level assumes that all neural activity within the high-d is possible. However, this chance level is too loose, given that the authors find that the neural activity resides in a low-d space. One way to address this is to find the overall dimensionality (N) of the population activity across tasks. Then, sample the 12-d subspaces from an N-dimensional space.

4) In Fig 4b, chance correlations are computed by shuffling the time points, which is again too loose. Instead, one should compute the chance level based on a random walk (with temporal smoothness matched to the original data) or some other temporally-correlated generated data.

5) The authors imply throughout the manuscript that the neural modes have a causal influence on movements (e.g., page 3 "motor cortex generates different motor behaviors through the flexible activations...of fixed neural modes"; page 15 "their role in generating these target-related EMG modes"). This seems like an overly strong interpretation of the neural modes. Based on the evidence shown in the paper, the neural modes are merely statistical characterizations of the population activity.

6) Supp Fig 9d: regarding the finding that target-related EMG is best predicted by target-related neural modes. Is there any way in which this would not have been true? In other words, the authors should make clear whether this is a finding or a sanity check.

Minor comments:

- page 9: "CCA is analogous...contain them." This statement is not entirely true, since CCA does not have any concept of time. It would be clearer to say that CCA identifies dimensions in two sets of data in which there is a correspondence between pairs of data points in each data set.

- page 10: "These results suggest an intriguing possibility:....motor output." To help the reader understand the significance of this statement, it would be helpful to state the alternative. In other words, what would the data look like if this weren't true?

- Are the neural modes with small principal angles in Fig. 3a related to the neural modes with large canonical correlations in Fig. 4b?

- Methods: were the different tasks interleaved in the same recording sessions?

- Supp Fig 3a is hard to read.

Responses to reviewers' comments

We thank all three reviewers for their detailed and thoughtful comments on our paper. The material and analyses added in response to their questions has helped strengthen our manuscript considerably. We have tried to carefully address each comment, which has resulted in several new analyses and a heavily revised paper. As consequence, this updated version has taken us a long time to produce, for which we apologize. In the revised manuscript, any relevant changes are highlighted in red. Below follows a comment-by-comment response to the Reviewers' reports.

Reviewer #1

This is a potentially interesting paper that attempts to compare neural manifolds across a range of tasks. Overall, it is an important area of investigation. As outlined below, however, there are several issues that limit enthusiasm. The primary concern is that multiple tasks were used and multiple related analytical approaches were employed without an apparently clear unifying framework.

We thank the reviewer for their interest in our work, and for the careful and useful comments on the manuscript.

Regarding the unifying theoretical framework, all the analyses are based on the idea that neural function is built on coordinated patterns of covariation across neurons (the *neural modes*) rather than on the independent modulation of single neurons¹. The collection of neural modes defines what is often called a *neural manifold*¹⁻³. In this view, the activity of each single neuron is one of the many different one-dimensional projections of the population activity^{4,5}. The analytical approaches that we employed were chosen to address fundamentally different ways in which the neural modes could change across tasks: their geometric structure, and their activation dynamics, the *latent activity*. We have incorporated a better description of this unifying framework and the logic underlying our analyses in the Introduction and the Results.

There is mention that the different methods may be similar, it would help to better explain the fundamental approach of using multiple yet related methods.

We thank the reviewer for motivating us to clarify our methods. We have restructured the paper so as to clarify our unifying hypothesis and how the analyses complement each other.

This is particularly challenging as multiple tasks are used in only a subset of animals. Ideally, the same set of tasks would have been used; it would then be easier to compare single neural responses and the population dynamics.

While this would have been ideal, the data were collected at different times or from different monkeys. This was the available data used for our analyses. Nonetheless, the two sets of tasks reinforce each other well. The set of wrist tasks allowed us to dissociate the movement from the forces exerted by the monkeys, a classic approach in studying motor control. The second set of tasks required more complex, less constrained behaviors involving reaching and grasping. The analyses allowed us to reach the same conclusions for both sets of tasks: that the neural manifolds for the different tasks in each set were well aligned, and that specific components in their latent activity were also preserved. These similarities were significantly larger than those observed for individual neurons or EMGs themselves, as illustrated by new analyses (Fig. 5; Suppl. Fig. 7g).

It would also better motivate exactly what the population dynamics represent relative to individual neurons. While this is not the case here, it would help to frame the manuscript from that perspective

As noted above, we view the activity of each single neuron as a one-dimensional projection of the population activity^{1,4,5}. This framework is now presented in the Introduction and other parts of the paper, and illustrated in Fig. 1c.

1. It is important to show that the single units (criteria used?) show similar results to the MU data. For example, it is quite possible that MUs appear to show common modes only because they are MU. If only SUs are used, is there common variance?

This question is thoroughly addressed in elegant recent work from the Shenoy and Ganguli groups⁶. They use experimental data from three previous studies⁷⁻⁹ as well as simulated data to show that the orientation of the neural manifold and the latent activity within it do not change when using single neurons or multi-units. We have now cited their paper and described why using sorted neurons or multi-units will not change our results. In addition, the robustness of both manifold orientation and its latent activity against unit choice is demonstrated by our own unit-dropping analysis, see Suppl. Fig. 3.

2. Better justification for 12 dimensions is important. It is surprising that only 60% of the variance is accounted for. Would this result hold for higher variance accounted for?

Although twelve dimensions did account for at least 60% of the neural variance across all datasets, the average variance accounted for (VAF) was $73 \pm 7\%$ (mean \pm s.d. over all monkeys, sessions, and tasks; see Suppl. Figure 3c). This average value is the one now highlighted in the paper. It must be noted that we computed the VAF based on concatenated trials; trial-averaging, as is commonly done in many related studies, would have increased the task-specific VAF considerably.

In addition, the results hold for higher manifold dimensionalities and higher VAF. Following the reviewers' suggestion, we have added a new figure (Suppl. Fig. 5) showing that the main results also hold for dimensionalities 8 and 15.

3. What does PCA of the EMG show? What is the variance accounted for relative to number of components?

The variance accounted for by a given number of *EMG modes* (the principal components) depends on the task: more EMG modes are needed to explain the same percentage of EMG variance as tasks require more co-activation patterns. Across all wrist datasets, 3.2 ± 0.8 EMG modes explained $\geq 95\%$ of the total EMG variance (new Suppl. Fig. 1d). Compared to the neural data, quite fewer EMG modes account for a larger percentage of the total variance.

4. Figure 1 is confusing given that multiple tasks are used. I would recommend making a more complete Figure 1 that can account for all the tasks used. It would help to also show the types of actions and how they might relate (e.g. isometric only vs wrist vs wrist/grasp).

We presume the reviewer is referring to Fig. 2, which shows example neural activity for the tasks. Following the suggestion, we have made a more complete version of this figure, which now also shows either the kinematics or force for the wrist tasks. For the reach-to-grasp tasks we did not measure limb kinematics.

5. Figure 1g. This could be the result of lumping in multiple conditions. What about comparing more similar vs more different conditions?

We calculated the distributions for the different pairs of tasks individually, and as there was little difference between them, we then combined the results (Response Fig. 1). To make this point more clearly, we have replaced what is now Fig. 2i in the paper with this updated version that shows the distributions for each set of tasks overlaid on top of each other.

Response Figure 1 Correlations between the activity pattern of each unit across each pair of tasks from either wrist (orange) or reach-to-grasp datasets (violet). Data pooled over all units, task comparisons, sessions, and monkeys within each dataset; top error bar: mean \pm s.d. correlation. Data points: wrist tasks: 1,476; reach-to-grasp tasks: 376; this difference is due to by having more tasks in the wrist sessions than the reach-to-grasp sessions, which increased the number of pairwise correlations.

6. Do the result hold if comparing 'more similar' vs 'more different' actions. In other words, comparison of two wrist based tasks vs a wrist and wrist/finger task. Again, this might help identify how the subspaces relate to the nature of task itself. As mentioned in the manuscript, comparisons of rodent walking and reaching show evidence of orthogonality.

We agree with the reviewer that this is a very interesting question, one that merits further exploration. Unfortunately, we could not compare wrist and reach-to-grasp tasks because our datasets are from different monkeys or different hemispheres. We have clarified this limitation in the Methods, and speculate in the Discussion on what might happen if we could have performed this comparison.

7. When using PCA, what were the distribution of weights in the PCs? Are the units with the highest weights similar across task (i.e. correlation of activity, this is related to point #5)?

Suppl. Fig. 3d in the original paper (reproduced as Response Fig. 2a) shows that the weights onto the leading neural modes define a narrow distribution centered slightly above 0. This analysis excludes outlier units with significantly higher weights and establishes that modes reflect population-wide patterns^{5,10}.

Moreover, we have now tested whether high-weight units, i.e., those with a weight within the first quartile of the weight distribution, displayed similar activity across tasks. We examined the relationship between their weights in different tasks. If the high-weight units were similar across tasks, the weights of those units in any two tasks would be strongly correlated. However, this correlation was very low, $r = 0.14$ (Response Fig. 2b). Thus, high-weight units were not similar across tasks, and the neural modes are true population-wide patterns.

Response Figure 2 (a) Distribution of neural unit weights onto the leading 12 neural modes, across all neural units for each task from each session and monkey (each shown in a different color). Inset: histogram summarizing all the data (same units as main figure in the panel; error bar: mean \pm SD). **(b)** Relationship between the weight of each unit with a high-weight in one task and its weight during another task. Data pooled across all task comparisons, sessions and monkeys. Line shows linear fit.

8. The analysis of angles between modes requires additional work. The manuscript uses random data to determine “small values. However, I could imagine that a more realistic dataset could result in a different conclusion. Could sufficiently scrambled neural data be used? The recent Cunningham lab (Nature Neuroscience) work could offer a strategy where only higher order statistics are perturbed (e.g. preserving the pairwise correlation matrix).

We thank all three reviewers for motivating us to design a more principled control. We have now implemented the tensor maximum entropy (TME) control proposed by the Cunningham lab¹¹. For each task, we generated surrogate datasets with the same covariance over time and the same covariance across targets as the experimental data¹¹. Since our alignment comparison between manifolds tests whether the covariance across neurons is similar for different tasks, the surrogate datasets did not preserve covariance across neurons; this would, by definition, have yielded the same principal angles as for the actual data.

Response Fig. 3a shows the principal angles between surrogate TME datasets for two representative sessions. The leading principal angles were significantly smaller ($P < 0.001$) than what would be expected based on the surrogate data, a result that held across all monkeys, sessions, and tasks (Response Fig. 3b).

We have also implemented an alternative test for the similarity of neural manifolds across tasks. If the manifolds from tasks A and B were well aligned, the projection of the neural data from task A onto the manifold from task B should account for a similar amount of variance as the projection of this data its own manifold¹². As shown in Response Fig. 3c, the cross-task manifold variance amounted to as much as $\sim 85\%$ of the within-task manifold variance.

Response Figure 3 Assessing the significance of the alignment between pairs of manifolds. **(a)** Principal angles for one session of reaching and grasping tasks from Monkey T (left) and for one session of wrist tasks from Monkey J (right). Each pairwise

comparison is shown as one colored trace (see legend). Leading principal angles were far below the $P=0.001$ significance level (dashed gray line) obtained using the maximum entropy method. **(b)** Number of neural modes for which all principal angles were significantly small across all monkeys, sessions, and task pairs. **(c)** Ratio of the neural variance accounted for (VAF) when projecting the neural data from one task onto the neural manifold from another task, to the VAF when projecting the same data onto its original manifold (black); data pooled over all monkeys, sessions, and task pairs. We compared this distribution to a control based on the 99.9th percentile of the distribution of VAF ratios when projecting neural data onto randomly generated manifolds (grey); the *** denote $P \sim 0$ (two-sided Wilcoxon rank sum test).

9. For CCA analysis, what is the VAF for the first 3 modes?

Manifold projections onto the three leading directions found using CCA explained $20.0 \pm 10.2\%$, $11.9 \pm 6.9\%$ and $5.9 \pm 2.4\%$ of the total neural variance, respectively (data pooled over all monkeys, sessions, and task comparisons). Response Fig. 4 shows the cumulative VAF; we have added this analysis as new Suppl. Fig. 7f.

Response Figure 4 Cumulative variance accounted for by the neural mode dynamics after projecting them along the new manifold directions identified with CCA. Grey traces: all comparisons across tasks, monkeys, and sessions; black solid and dashed lines: mean \pm s.d.

Minor.

1. The manuscript used fairly dramatic language. For example, the word critical is used often. It is not clear, at least to me, that all of these are warranted. Consider reducing bombast...

We have taken this recommendation into account and edited the manuscript accordingly.

2. Page 22: Reference 50/43 appear to be incorrect. These refer to methods used in this manuscript.

We had cited those papers because they describe our lab's standard surgical and post-operative care methods. In the revised version we have only left Ref. 43, as it contains sufficient detail.

3. Supp Figure 2. Scale bars are not shown. What about raw waveforms (i.e. not only the mean). ISI distribution?

We have now added scale bars to Suppl. Fig. 2. The raw waveforms look very similar across tasks, as exemplified for some units in Response Fig. 5. We did not include the ISI distributions for the neural units across different tasks because they look quite different across different tasks, a difference that does not reflect changes in unit identity.

Response Figure 5 Examples of action potential waveforms (100 randomly chosen threshold crossings per unit and task) for the datasets of wrist tasks **(a)** and reach-to-grasp tasks **(b)** shown in Fig. 2 in the paper. Scale bar: 100 μ V.

4. Page 24. How was the firing rate smoothed?

As described in the Methods, we binned the firing rates (bin size: 20 ms) and convolved a Gaussian kernel (s.d.: 50 ms). This combination of parameters is quite standard in the field (see, e.g., Ref. 13). The results did not change if we modified these parameters within a reasonable range.

Reviewer #2 (Remarks to the Author):

This paper hypothesizes the control of a variety of movements as arising from population neural activity in the primary motor cortex (M1) that can be well explained by a reduced set of neural ‘modes’. The idea of a reduced-dimension neural manifold to represent the neural activity has been shown to exist in different settings, with an insightful review [1] by a subset of the authors containing some of the key references. The key question being addressed in this paper is: how closely aligned are the neural manifolds in a variety of different tasks, i.e. how far away from orthogonal are they? Indeed, if the neural manifolds perfectly overlap during a wide variety of tasks, and the only difference between the tasks is the dynamics of the activity within these manifolds, it would be an extremely insightful finding and may lead to some important discoveries about the structure of connections in M1. The analyses performed in this paper seem to suggest that the manifolds are indeed aligned at a significant level across a variety of tasks. The second question being addressed in this paper is: does the neural activity evolve similarly or differently in these lower dimensional neural manifolds, across tasks? The authors use two analysis techniques to answer this question. The first analysis method (CCA) leads to the result that the neural activity evolves in a fairly similar manner across tasks, which is subsequently clarified by a second analysis method (dPCA) that finds a substantial component of the activity to evolve in a task-independent manner, while the rest to evolve in a task-dependent manner. Lastly, this paper addresses the question: can the muscle activity be explained or decoded by the lower dimensional neural manifold? Indeed, this is shown to be possible to some extent in these tasks. Overall, the main finding of Gallego et al. is the similarity of the neural manifolds during a variety of tasks, with the additional analyses serving to complement the main hypothesis and add context to the datasets. (...)

We thank the reviewer for a positive reaction to our work, and for the insightful comments. We have addressed them, and thus considerably strengthened our paper.

Major Concerns: In my view, this paper cannot be published until the following concerns are addressed.

1) Since the crux of the paper lies in the presence of small principal angles between the neural manifolds of different tasks, this analysis requires extra care. As the authors note, the threshold angles between 12D manifolds increased with the dimensionality of the neural space (Suppl. Fig. 5b). However, the fact that the thresholds increase considerably with n the number of neurons recorded is fundamentally problematic. This increase can be conceptualized as the following: the probability that two 2D vectors randomly sampled from a 3D space are orthogonal is lower than the probability that two 2D vectors randomly sampled from a 4D space are orthogonal. An alternative suggestion is to rely on permutations to compute the significance thresholds, as in [3], or, better yet, to draw random subspaces biased to the data covariance structure, as in [4] (see Supplementary Note 3 in [4]). This analysis will make the results of this paper much stronger.

The reviewer is correct that the probability that two randomly sampled subspaces are orthogonal increases with the dimensionality of the space in which the subspaces are embedded. To control for this property, our original control involved generating random pairs of 12-dimensional subspaces (manifolds) embedded in a space of the same dimensionality as that of the neural data. We hope that the new rendition (now, Suppl. Fig. 4a,b) clarifies this issue.

In addition, and following the suggestion of all three reviewers, we have replaced randomly generated control data by control data generated using the tensor maximum entropy (TME) method, one of the “neural population controls” proposed in Ref. 11. (See response to Reviewer 1 Comment 8 above). We have also performed an additional analysis that compares manifolds from different tasks based on the fraction of variance accounted for (VAF) when projecting data from one task onto the manifold from a different task. This analysis is similar to the comparison of preparatory and movement subspaces in [4].

It is also key to have a discussion around the dependence of this threshold and the subsequent results on m , the number of PCs chosen, here 12. One suggestion is to show the analysis for varying m in a reasonable range, say 5 to 15.

This is an excellent point. Following the suggestion of both Reviewers 1 and 2, we have added a new figure (Suppl. Fig. 5) that shows our results to be qualitatively unchanged for different sizes of manifolds, $m=8$ and 15.

2) An important question being asked here is the similarity of neural manifolds between a variety of tasks. That being said, the authors only compute the principal angles and perform the correlation analysis on pairs of ‘similar’ tasks. It would benefit the hypothesis if the principal angles between the three dimensional (“ball” and

“grip” tasks) and the one dimensional tasks (“iso”, “mov”, “spr” tasks) are also small. It would also be interesting if these principal angles are larger than the principal angles between just the one dimensional tasks. This analysis may only be possible for one monkey (monkey C), but would go a long way towards showing that a larger variety of tasks have aligned neural manifolds. Without this, the reader is left wondering why this analysis was not performed.

We agree with the reviewer that comparing the wrist and reach-to-grasp tasks would have been interesting. Unfortunately, the data available at this point made a direct comparison impossible, as noted above (see response to Reviewer 1 Comment 6).

3) It is key to have a thorough discussion of the variety in the different tasks in the context of past studies. For example, how different are the kinematics across tasks in this study, as compared to other motor studies mentioned in the references? The larger the variety in kinematics arising from the same or aligned neural manifolds, the more surprising and thus impactful are the results. The authors have shown the large variety in Suppl Figures 1 and 2, but a discussion of this variety in the context of past experimental studies is lacking. It is a strength of this paper to have a large variety of different tasks – this point can be stressed further throughout this paper.

We thank the reviewer for pointing out that we had not discussed these topics with enough detail. We have now added a new section to the Discussion, “Relation to previous single neuron representational studies”. Prior studies, as well as our results, show complex changes in single neuron activity and the variables that they encode across upper limb tasks (e.g., Refs. 14–18). Yet, we have found that for the tasks we analyzed, their neural manifolds are well aligned, and that the activity of a subset of the modes that span these manifolds is also preserved. These similarities are greater than what would be expected based on the across-task similarities of neural units or muscle activity (new Fig. 5 and Suppl. Fig. 7g).

Moderate Concerns: The following points are very strongly suggested before publishing this paper.

1) The three-dimensional tasks, i.e. “ball” and “grip” are significantly more complex than the one and two dimensional tasks. A discussion on the difference in the activation dynamics in the neural manifold or differences in the nature of the neural manifold itself may be insightful. It would be beneficial to add an analysis on the amount of variance captured by the PCs calculated on the one dimensional task to the neural space occupied by the three dimensional tasks.

We agree that this is a very interesting topic for investigation. However, as stated in the response to Major Comment 2, direct comparison between the reaching and wrist tasks is difficult. Instead, we have used the 1-D and 2-D wrist tasks to begin to understand these relationships, as we were able to compare data across these tasks while performed by the same monkey and recorded using the same electrode array.

For monkey J, fewer modes were required to explain a given amount of variance for the 1-D elastic-loaded movement task (“spr”) than for any other wrist task (Response Fig. 6a). Unexpectedly, there was not a clear difference even between the 1-D and 2-D versions of the isometric task. In contrast, for monkey C there was no clear separation between 1-D tasks (Response Fig. 6b). We have not included these new analyses in the paper because the results were inconclusive.

Response Figure 6 Neural variance accounted for as function of the number of neural modes for all wrist task. Data taken during the three wrist sessions of monkey J (a), and the three wrist sessions of monkey C (b). The lower VAF for monkey C is due to the lower SNR of the recorded data.

4) The analysis regarding the correlations in the ‘neural mode dynamics’ between different tasks, while interesting, paints an unfinished picture, which is subsequently clarified by the dPCA analysis. The across-task CCs as reported by the authors are not extremely high (though comparable to within-task CCs), as expected since a part of the neural activity does actually differ across tasks (otherwise task discrimination based on

neural activity would be impossible). This point is clarified by the dPCA analysis, which shows that a significant part of the neural activity can be explained by task-independent effects, whereas the rest from task-dependent effects, while quantifying these effects. The CC analysis presumably captured the task-independent effects. However, it is unclear what additional information the CC analysis presents (a better motivation of the CC analysis might help clarify this point). It is clear that the dPCA analysis could not be carried out for a subset of the tasks, and thus a CC analysis would be warranted for these – though it is a suggestion that this may be moved to supplementary material.

We thank the reviewer for this suggestion. We debated these two alternative approaches to presenting our results when writing the previous version of the paper. In the current version we have relegated the CCA to the Supplementary Material. The logical flow of the argument has improved.

5) As mentioned in the discussion (page 18) and corroborated by Reference 60 in the text, it is possible that the results in the paper are expected if I believe that the task-independent signal is the ‘largest’ signal in the neural activity. A discussion on this point is warranted earlier and much more prominently. Reference [5] may be relevant.

The reviewer brings up a good point that we should have discussed more clearly. In agreement with Ref. 60, we indeed do find several neural modes whose activity is the same for all the tasks and conditions, the “time-related” modes. Although Kaufman et al. were the first to report a target-independent neural mode¹⁹, we have found that this set of neural modes is not only target, but also task independent. Moreover, we also found another set of task-independent modes, the “target-related” modes, whose activity covaries with the motor action, as indicated by the location of the target. These modes explain ~30% of the total neural variance, an amount similar to that explained by the time-related modes (Figure 4c). We have described this result in the Discussion, in a new section titled “Interpreting neural modes with task-independent activity”.

6) Although the dPCA analysis lends well to interpretability, more information could be provided on how well the dPCs capture the per-task variance. Moreover, a discussion on whether the neural manifold identified by dPCA is close to those identified by the separate PCAs per task would be helpful.

The reviewer raises two interesting questions. The 12-D dPCA manifolds that spanned all the wrist tasks for a given session, the “multi-task dPCA manifolds,” explained as much as $87 \pm 5\%$ of the total task specific variance of the trial-averaged data (Response Fig. 7a, now Suppl. Fig. 6f). This amount of VAF was very close to what PCA yielded: $89.5 \pm 0\%$. The dPCA manifolds captured task-specific neural data almost as well as the PCA manifolds did.

In addition, a comparison of the latent activity (i.e., the dynamics of the neural modes) for each task-specific PCA manifold to the latent activity within the multi-task dPCA manifold shows appreciable similarities (Response Fig. 7b).

Response Figure 7 Relationship between the task-specific PCA neural manifolds and the multi-task dPCA neural manifolds. **(a)** Per-task variance spanned by the multi-task dPCA manifolds; bars: mean + s.d.; data pooled across tasks and sessions for each monkey separately. As required by dPCA, these calculations were based on trial-averaged data and results in larger VAF values. **(b)** CCA comparison of the dynamics of the neural modes found with PCA and dPCA; data for all wrist tasks performed by both monkeys (thin pink traces); solid and dashed purple lines: mean \pm s.d.

7) Since the R^2 values for the target-related neural modes to the entire EMG activity in different tasks are modest, it is my opinion that the term “stable control of varied behavior” is perhaps an overstatement of the results. The target-related modes of the EMG can be decoded by the target-related neural modes fairly well, but the EMG activity contains other modes that the target-related neural manifold is apparently not entirely successful at decoding, which itself may be the cause of ‘varied’ behavior. Thus, the last sentence in the results section is accurate: the authors have “identified a stable, task-independent component in the mapping from specific neural modes to specific EMG modes”. The title of the paper may not accurately reflect this result.

We agree with the reviewer that a claim of “stable control of varied behavior” is not a fully accurate representation of our results. We have therefore changed it to “Stable components in motor cortical control of behavior.”

Minor Concerns: The following points are strongly suggested, but should not change the results or content of the paper too much.

1) The abstract should be modified to include a specific scientific question and the results of the paper in the context of other findings in the motor community.

We have rewritten the abstract to address this concern.

2) Reference [5] may be relevant, along with the references 14 and 15 already listed in the paper in the second paragraph of the introduction (page 2).

We have added a citation to this paper.

3) It would be helpful to compare the results in the paper to existing results in the motor community in the last paragraph of the Introduction section.

Done.

4) In the CCA Methods section, the authors mention taking the average between the upper bounds of the CC distribution of two tasks (page 28). It is suggested to take the maximum instead.

We have implemented this change as requested (Suppl. Fig. 7b,d). We note that this does not change the results.

5) The authors should detail how they equalized the number of trials across all tasks within the corresponding session, as mentioned on page 27 and page 30.

We took the first k trials for each task and target, with k being the minimum number of successful trials across all targets and tasks in the session. We have described this procedure more clearly in the Methods.

6) The bootstrapping method to assess significance in CCA analysis should be clarified and detailed (page 28).

To test whether the neural mode dynamics were preserved across tasks, we originally randomized in time the latent activity for one of the tasks (for all modes simultaneously) and used CCA to compare these surrogate data to the data from a second task. We repeated this 10,000 times for each task comparison, and took the 99.9th percentile of the resultant distribution of CCs as a significance threshold. This method allowed us to test the stability of the latent activity while keeping the neural covariance structure unaltered.

Following the suggestion by Reviewer 3 (Comment 4), we have now modified the CCA control so as to have surrogate data that preserve the smoothness of the original data. We note that this does not change our findings of significant stability in latent activity.

7) It may be helpful to show or mention whether the dPCA results depend significantly on m .

The dPCA results did not depend on the chosen manifold dimensionality. New Suppl. Fig. 5b shows that the features of latent activity shown for $m = 12$ in Fig. 4 of the paper are also present for $m = 8$ and 15.

8) The plots in Suppl Figure 3a warrant further discussion. For the wrist tasks, we see a relatively large number of neurons with high PC weights on the tail end of the neural modes, which is not seen in the reach-to-grasp tasks. It could be insightful to investigate why.

The reviewer makes an interesting point. We have run new analyses to understand the potential role of these high-weight units. First, we quantified whether the percentage of high-weight units was in general greater for the wrist tasks than the reach-to-grasp tasks, as suggested in the example sessions that the reviewer indicates. We focused on the leading 12 modes, as we are interested in those modes that explain a large part of the population variance and were thus selected to specify the neural manifold. When we set a threshold onto the magnitude of the weight of the units across all datasets (the 1st percentile across all datasets yielded a threshold of 0.48 for considering a weight as high), more units did indeed have a high weight during the wrist tasks than during the reach-to-grasp tasks (13 % vs. 7 %).

We then sought to understand if this difference indicates that during wrist tasks the leading neural modes were less reflective of population-wide activity patterns. If this were the case, we would expect certain units to be more highly weighted across wrist tasks than reach-to-grasp tasks. This was not the case (Response Fig. 9).

Response Figure 9 The relationship between the weight of each unit with a high weight in one task and its weight in another task was statistically indistinguishable when making a comparison between wrist tasks (black) and between the two reach-to-grasp tasks (red). Data pooled across all task comparisons, sessions, and monkeys, separately for each dataset. Line shows linear fits (same color code).

9) The authors use a specific static nonlinearity that seems arbitrarily chosen to decode the neural mode dynamics into the EMG activity. Did the authors try polynomials of different order, or different classes of nonlinearities? Motivating the second order polynomial is suggested. Moreover, the decoder is listed as linear in several other places in the paper, despite the presence of a nonlinearity.

The decoder is indeed a linear-nonlinear Wiener cascade, and we have now named it properly as “a standard Wiener cascade decoder” including pertinent references.

Based on previous work from our group²⁰⁻²², we tried polynomials of order two and three, and found the results barely changed: there was less than a 2% improvement when using a third order polynomial. Thus, we opted for a second order polynomial to keep the model parameters to a minimum.

10) Although the normalized R^2 values of the EMG predictions are listed, as calculated on page 31, the R^2 values of the EMG predictions based on all 12 neural mode dynamics are not listed, which would be helpful as a comparison.

The R^2 of the cross-validated EMG predictions using the activity of all 12 dPCA neural modes as inputs was 0.53 ± 0.21 . We have added this result to the section where we described the decoding methods.

Readability Concerns:

1) The authors seem to use ‘activation dynamics’, ‘time varying activation’, and ‘neural mode dynamics’ interchangeably. For the sake of readability, it is suggested that the authors stick to one of these terms, or clarify the differences between them.

We have now rephrased all instances to the common term^{1,13,23,24} “latent activity,” which we define in the Introduction as the neural mode dynamics, i.e. the population dynamics within the manifold.

2) Modify the first sentence of the second paragraph on page 3 to clarify that the hypothesis is that the same or aligned neural modes are activated during different tasks.

Done.

3) In Figure 1, is u_i the neural mode or the activation dynamics? The text would suggest neural mode. Also clarify this in the caption for Figure 1.

It is the neural mode, as the reviewer says. We have corrected the caption accordingly.

4) The second sentence of the first paragraph of the Discussion section reads awkwardly (page 16). "Critically, several of these studies..."

We have now rephrased this sentence.

5) It is not certain that Reference 11 on page 18 is correctly placed.

Sussillo et al. made this statement referring to any pattern generating network (as M1) in the Discussion of their paper²⁵. We have kept this reference and added Refs. 1, 26, and 27, where similar statements are made about motor cortical population activity.

6) There is a typo in first line of the third paragraph on page 27. hose → those.

Changed.

References:

- [1] Gallego, J. A. et al. Neural manifolds for the control of movement. *Neuron* (2017).
- [2] Churchland, M. M. et al. Neural population dynamics during reaching. *Nature* (2012).
- [3] Miri, A. et al. Behaviorally selective engagement of short-latency effector pathways by motor cortex. *Neuron* (2017).
- [4] Elsayed, et al. Reorganization between preparatory and movement population responses in motor cortex. *Nature Comms.* (2016).
- [5] Elsayed et al. Structure in neural population recordings: an expected byproduct of simpler phenomena? *Nature Neurosci.* (2017).

Reviewer #3 (Remarks to the Author):

In this work, the activity of tens of neurons is recorded in primary motor cortex (M1) while monkeys perform different manual tasks, including wrist isometric and movement tasks and grip tasks. The authors find that the neural modes identified from different tasks are similar, both in terms of the contribution of different neurons to the neural modes (PCA analysis) as well as the temporal dynamics (CCA analysis). Furthermore, they find task-independent neural modes (dPCA analysis) and relate them to muscle activity (EMG).

The data are extremely rich with the same neural data recorded during these different tasks. No previous study has compared M1 population activity recorded in as many different tasks. Furthermore, the authors take a modern view on M1 by characterizing the neural activity in terms of neural modes, rather than categorizing single neurons. For these reasons, this line of work has great potential. However, I have several substantial concerns (see details below).

We thank the reviewer for the appreciation of our work and for the insightful comments. We have implemented the recommendations, which helped greatly improve the manuscript.

Major comments:

1) It is unclear to what extent these results are simply a result of the EMG being similar for these different tasks. Let's say that there is a fixed relationship between M1 and muscles. Given the similarity of the EMG across tasks, would we expect to see the degree of similarity of the M1 activity across tasks that the authors report? In other words, have the authors found anything interesting in the neural activity beyond what can be deduced from EMG alone? To address these questions, I have several concrete suggestions:

1a) If the analysis in Fig 3 is repeated on EMG, would we see a similar result?

Following this suggestion, we have applied the principal angle analysis to the *EMG manifolds*, the subspaces defined by the dominant muscle co-activation patterns or *muscle synergies*²⁸⁻³¹ for each task. Response Fig. 10 shows that the structure of the EMG modes is considerably less similar across tasks than the structure of the neural modes; in most cases (65%) either no or at most one principal angle between EMG manifolds was significantly small ($P < 0.001$). This is in sharp contrast starkly with the principal angles between neural manifolds; in all cases at least 10 of 12 possible neural modes were significantly small based on our control ($P < 0.001$). This result, together with the results that address the next two comments, show that the observed similarities between neural modes cannot simply be explained by similarities in muscle activity. We have incorporated Response Fig. 10 into a new Fig. 5.

Response Figure 10 Comparison of the EMG manifolds and neural manifolds across tasks using principal angles. The histogram summarizes the percentage of principal angles between neural manifolds and EMG manifolds that are below the TME significance threshold ($P < 0.001$) across all task comparisons, sessions and monkeys ($n=29$). The X-axis is normalized to the percentage of total neural or EMG manifold dimensions (neural manifold: 12; EMG manifold: 4).

1b) Supp Fig 7c is the only analysis which shows that there may be something interesting in the neural modes beyond what can be deduced from EMG. However, this result may be a trivial consequence of the fact that more

neural modes (12) than EMG modes (7) are used in this analysis. Does the same result hold with an equal number of neural and EMG modes? Also, canonical correlations (like Pearson correlation) is sensitive to the amount of noise in the data. Is the amount of noise in the trial-averaged neural and EMG data similar?

The reason we used different numbers of neural and EMG modes is that considerably more neural modes were needed to explain a reasonable fraction of the variance in the neural data. We have now repeated this analysis making the dimensionality of the neural manifold equal to the number of EMGs (from 5 to 11, depending on the session; on average: 8.2 ± 2.8), and the results hold: the similarities in the dynamics of the neural modes (now called *latent activity*, following a recommendation from Reviewer 2) are greater than the similarities in the EMGs (Response Fig. 11a).

Regarding the inter-trial noise, we have estimated the signal-to-noise ratio (SNR) of the latent activity and the EMGs across all datasets. The SNR of the EMGs is significantly higher than that of the neural modes ($P \sim 0$, two-sided Wilcoxon rank sum test; Response Fig. 11b). Thus, the dissimilarity between EMGs is not due to lower SNRs.

Response Figure 11 Additional comparisons between latent neural activity and the EMGs. **(a)** Ratio of the across-task CC between latent activities to the across-task CC between EMGs, when making the neural manifold dimensionality equal to the number of EMGs. Data pooled over all tasks, sessions and monkeys. The fact that most values are >1 indicates that the latent activities were more similar than the corresponding EMGs. **(b)** Signal-to-noise ratio (SNR) of the latent activity and the EMGs. The SNR of the EMGs was significantly higher ($P \sim 0$) than that of the latent activity; data pooled over all tasks, wrist sessions and monkeys.

1c) Construct a model in which M1 and EMG has a fixed relationship, and give the model the recorded EMG. Then show that the results found in this paper about M1 do not fall out of this model.

We thank the reviewer for this suggestion. We have implemented it, and found that the similarity in the actual neural activity is significantly larger than what the model predicts.

Our model, largely based on that in Ref. 32, is as follows (Response Fig. 12a): each neural unit j was simulated as a Poisson point process with a time-dependent mean: $\lambda_j(t) = a_j + \sum_i b_{ji} \mathcal{E}_i(t) + \eta_j(t)$, with $0 \leq a \leq 0.1$ and $-1 \leq b \leq 1$. Here \mathcal{E}_i is the EMG of the i th recorded muscle, and η_j is random Gaussian noise drawn from a normal distribution $\mathcal{N}(0,0.05)$ to match the SNR of the neural recordings. For each task, we simulated as many units as experimentally recorded. We scaled their binned spike trains so that the mean firing rates matched those of the actual neural data. The simulated data also captured second order statistics of the actual data: the variance explained by the neural modes of the simulated and actual data followed a similar trend (Response Fig. 12b). Finally, we used CCA to compare the simulated latent activity across every pair of wrist tasks that monkeys performed in each session. The canonical correlations (CCs) of the simulated data were significantly smaller than those of the actual neural data ($P \sim 0$, two-sided Wilcoxon rank sum test; see data in Response Fig. 12c). Therefore, our results cannot be trivially explained as similarities in neural manifold activity being a representation of similarities in the ongoing muscle activity. We have incorporated this result to the paper as part of the new Fig. 5.

Response Figure 12 A model that assumes that neural activity is primarily a representation of the muscle activity does not explain the experimental results. **(a)** Model schematic: neural spiking results from weighted combinations of the recorded EMGs. **(b)** The cumulative variance accounted for (VAF) (the eigenvalue distributions) for the simulated and actual neural data are very similar. **(c)** The canonical correlations (CC) across tasks are significantly smaller for the simulated data. The observed larger CCs for the actual neural data do not follow from similarities in muscle activity across tasks.

1d) Given that ~70% of the EMG variance is task-independent (Supp Fig 9), is it surprising that ~60% of the neural variance is task-independent (Fig 5c)? The model suggested above could help bolster this claim.

The simulation results obtained with the model suggested by the reviewer indicate that the observed similarities in latent neural activity are not trivially explained by M1 activity merely representing ongoing muscle activity (Response Fig. 12c). The fact that the orientations of the EMG manifolds differ more across tasks than those of the neural manifolds (Response Fig. 10), together with the greater differences in EMGs than in latent neural activity (now Fig. 5b,c) provide further evidence in this regard.

The dPCA of the neural activity illuminates why the numbers the reviewer refers to are actually surprising: as shown in Figure 4c, ~50 % of the task-independent neural variance corresponds to modes whose dynamics do not depend on the specifics of the generated motor output, neither the task nor the target to be acquired. Only the remaining ~50% of task-independent neural variance, the target-related modes, corresponds to neural modes that are predictive of the EMGs. Thus, there is more modulation of task-independent components to neural activity than the modulation related to muscle activity.

2) One of the drawbacks of principal angles and canonical correlations is that they do not take into account the amount of variance in the data. In Fig 3a and 4b, the amount of variance in the data explained by the neural modes with low principal angles (Fig 3a) or high canonical correlations (Fig. 4b) should be indicated. If these neural modes explain large amounts of variance in the data, then the results can be interpreted as being more meaningful.

We thank the reviewer for pointing this out. We have performed this analysis and found that the modes associated with low principal angles or high canonical correlations (CCs) account for more variance than those corresponding to large angles or low CCs.

We have included the corresponding VAF values to the example traces in Fig. 4 (now Suppl. Fig. 7); we have also included the new Suppl. Figs. 4c and 7e, which show the VAF along the manifold directions associated with principal angles and CCA (Response Figs. 4 and 13).

Response Figure 13 Variance explained for along the manifold directions identified with the principal angles analysis. The figure shows all comparisons across tasks, monkeys, and datasets (grey traces), and the mean \pm s.d. (full and dashed black lines).

3) In Fig 3a, the chance level assumes that all neural activity within the high-d is possible. However, this chance level is too loose, given that the authors find that the neural activity resides in a low-d space. One way to address this is to find the overall dimensionality (N) of the population activity across tasks. Then, sample the 12-d subspaces from an N -dimensional space.

The control the reviewer describes is what we had done in the paper; we apologize if the original description was not clear. In the original version, the surrogate data for control was generated by sampling randomly oriented 12-dimensional manifolds within the n -dimensional neural space. In the current, revised version, we have followed the recommendation from all three reviewers, and generated the control manifolds by using the tensor maximum entropy (TME) method recently proposed in Ref. 11 (see response to Reviewer 1 Comment 8, and Response Fig. 3). We have also performed an additional analysis that compares manifolds for different tasks based on the amount of VAF when projecting data from one task onto the manifold for a different task.

4) In Fig 4b, chance correlations are computed by shuffling the time points, which is again too loose. Instead, one should compute the chance level based on a random walk (with temporal smoothness matched to the original data) or some other temporally-correlated generated data.

Our original control was designed to test for features in the neural covariance that had similar activity across tasks. The reviewer is correct in that this control produced surrogate datasets for which the temporal evolution within the manifold (i.e., the latent activity) was less smooth than for the original data. We now smooth the surrogate data with a Gaussian kernel with a width chosen to match their spectral content (s.d.=50 ms), after shuffling. Response Fig. 14 shows that although this new control slightly decreases the number of modes with well-preserved activity, it still identifies significant across-task similarities. We have updated the control and figures in the paper accordingly.

Response Figure 14 Number of neural modes with similar dynamics across tasks ($P < 0.001$), using our new control for comparison against actual CC values. Data pooled over all monkeys, sessions, and task comparisons.

5) The authors imply throughout the manuscript that the neural modes have a causal influence on movements (e.g., page 3 "motor cortex generates different motor behaviors through the flexible activations...of fixed neural modes"; page 15 "their role in generating these target-related EMG modes"). This seems like an overly strong interpretation of the neural modes. Based on the evidence shown in the paper, the neural modes are merely statistical characterizations of the population activity.

The reviewer is correct that our observations do not allow us to assert causality. Our original statements were motivated by the well-established causal influence of primary motor cortical neurons onto upper limb muscles. However, since our evidence is only correlative, we have rephrased these statements to mention that our evidence is indirect.

6) Supp Fig 9d: regarding the finding that target-related EMG is best predicted by target-related neural modes. Is there any way in which this would not have been true? In other words, the authors should make clear whether this is a finding or a sanity check.

The reviewer is correct: we performed this analysis to understand why we could predict EMGs for different tasks quite accurately based on only target-related but task-independent neural modes. As expected, the reason is the existence of a target-dependent but task-independent component in the EMGs.

Minor comments:

- page 9: "CCA is analogous...contain them." This statement is not entirely true, since CCA does not have any concept of time. It would be clearer to say that CCA identifies dimensions in two sets of data in which there is a correspondence between pairs of data points in each data set.

It is correct that CCA does not incorporate the notion of time, but just compares sets of pairs of corresponding data points, in this case the neural trajectories, embedded in a multi-dimensional space. We have rephrased the sentence in the paper to describe the analysis both accurately and intuitively.

- page 10: "These results suggest an intriguing possibility:....motor output." To help the reader understand the significance of this statement, it would be helpful to state the alternative. In other words, what would the data look like if this weren't true?

We have added an alternative possibility, that the brain generates movement by independently modulating the activity of single neurons.

- Are the neural modes with small principal angles in Fig. 3a related to the neural modes with large canonical correlations in Fig. 4b?

The modes with small principal angles and the modes with large CCs are related by construction, as they are defined by different directions within the same neural manifold. In other words, they both are linear combinations of one another and also of the original neural modes obtained by PCA.

- Methods: were the different tasks interleaved in the same recording sessions?

All the tasks we compared were performed in blocks in the same experimental session. We have clarified this in Methods.

- Supp Fig 3a is hard to read.

We have modified Suppl. Fig. 3a to make it clearer.

References

1. Gallego, J. A., Perich, M. G., Miller, L. E. & Solla, S. A. Neural Manifolds for the Control of Movement. *Neuron* **94**, 978–984 (2017).
2. Stopfer, M., Jayaraman, V. & Laurent, G. Intensity versus identity coding in an olfactory system. *Neuron* **39**, 991–1004 (2003).
3. Sadtler, P. T. *et al.* Neural constraints on learning. *Nature* **512**, 423–6 (2014).
4. Gao, P. *et al.* A theory of multineuronal dimensionality, dynamics and measurement. *bioRxiv* 214262 (2017). doi:10.1101/214262
5. Ganguli, S. & Sompolinsky, H. Compressed Sensing, Sparsity and neural data. *Annu. Rev. Neurosci.* **35**, 463–483 (2012).
6. Trautmann, E. M. *et al.* Accurate estimation of neural population dynamics without spike sorting. 1–42 (2017). doi:10.1101/229252
7. Ames, K. C., Ryu, S. I. & Shenoy, K. V. Neural dynamics of reaching following incorrect or absent motor preparation. *Neuron* **81**, 438–51 (2014).
8. Churchland, M. M. *et al.* Neural population dynamics during reaching. *Nature* (2012). doi:10.1038/nature11129
9. Kaufman, M. T., Churchland, M. M., Ryu, S. I. & Shenoy, K. V. Cortical activity in the null space: permitting preparation without movement. *Nat. Neurosci.* **17**, 440–8 (2014).
10. Gao, P. & Ganguli, S. On simplicity and complexity in the brave new world of large-scale neuroscience. *Curr. Opin. Neurobiol.* **32**, 148–155 (2015).
11. Elsayed, G. & Cunningham, J. P. Structure in neural population recordings: significant or epiphenomenal? *Nat Neurosci* **25**, 1–14 (2017).
12. Elsayed, G. F., Lara, A. H., Kaufman, M. T., Churchland, M. M. & Cunningham, J. P. Reorganization between preparatory and movement population responses in motor cortex. *Nat. Commun.* 13239 (2016). doi:10.1038/ncomms13239
13. Yu, B. M. *et al.* Gaussian-Process Factor Analysis for Low-Dimensional Single-Trial Analysis of Neural Population Activity. *J. Neurophysiol.* **102**, 614–635 (2009).
14. Kakei, S., Hoffman, D. S. & Strick, P. L. Muscle and movement representations in the primary motor cortex. *Science* **285**, 2136–9 (1999).
15. Kurtzer, I., Herter, T. M. & Scott, S. H. Random change in cortical load representation suggests distinct control of posture and movement. *Nat. Neurosci.* **8**, 498–504 (2005).
16. Morrow, M. M., Jordan, L. R. & Miller, L. E. Direct comparison of the task-dependent discharge of M1 in hand space and muscle space. *J. Neurophysiol.* **97**, 1786–98 (2007).
17. Hepp-Reymond, M. C., Kirkpatrick-Tanner, M., Gabernet, L., Qi, H. X. & Weber, B. Context-dependent force coding in motor and premotor cortical areas. *Exp. Brain Res.* **128**, 123–133 (1999).
18. Kalaska, J., Cohen, D., Hyde, M. & Prud'homme, M. A comparison of movement direction-related versus load direction-related activity in primate motor cortex, using a two-dimensional reaching task. *J. Neurosci.* 2080–2102 (1989).
19. Kaufman, M. T. *et al.* The Largest Response Component in the Motor Cortex Reflects Movement Timing but Not Movement Type. *eNeuro* **3**, (2016).
20. Pohlmeier, E. A., Solla, S. a, Perreault, E. J. & Miller, L. E. Prediction of upper limb muscle activity from motor cortical discharge during reaching. *J. Neural Eng.* **4**, 369–79 (2007).
21. Ethier, C., Oby, E. R., Bauman, M. J. & Miller, L. E. Restoration of grasp following paralysis through brain-controlled stimulation of muscles. *Nature* **485**, 368–71 (2012).
22. Oby, E. R., Ethier, C. & Miller, L. E. Movement representation in the primary motor cortex and its contribution to generalizable EMG predictions. *J. Neurophysiol.* **109**, 666–78 (2013).
23. Cunningham, J. P. & Yu, B. M. Dimensionality reduction for large-scale neural recordings. *Nat. Neurosci.* **17**, 1500–1509 (2014).
24. Roweis, S. & Ghahramani, Z. A Unifying Review of Linear Gaussian Models. *Neural Comput.* **11**, 305–345 (1999).
25. Sussillo, D., Churchland, M. M., Kaufman, M. T. & Shenoy, K. V. A neural network that finds a naturalistic solution for the production of muscle activity. *Nat. Neurosci.* **18**, 1025–33 (2015).
26. Sergio, L. E., Hamel-Pâquet, C. & Kalaska, J. F. Motor cortex neural correlates of output kinematics and kinetics during isometric-force and arm-reaching tasks. *J. Neurophysiol.* **94**, 2353–78 (2005).
27. Russo, A. A. *et al.* Motor Cortex Embeds Muscle-like Commands in an Untangled Population Response. *Neuron* 1–14 (2018). doi:10.1016/j.neuron.2018.01.004
28. d'Avella, A., Saltiel, P. & Bizzi, E. Combinations of muscle synergies in the construction of a natural motor behavior. *Nat. Neurosci.* **6**, 300–308 (2003).
29. Tresch, M. C. & Jarc, A. The case for and against muscle synergies. *Curr. Opin. Neurobiol.* **19**, 601–7 (2009).
30. Giszter, S. F. Motor primitives-new data and future questions. *Curr. Opin. Neurobiol.* **33**, 156–165 (2015).
31. Tresch, M. C., Cheung, V. C. K. & D'Avella, A. Matrix factorization algorithms for the identification of muscle synergies: evaluation on simulated and experimental data sets. *J. Neurophysiol.* **95**, 2199–2212 (2006).
32. Perich, M. G. & Miller, L. E. Altered tuning in primary motor cortex does not account for behavioral adaptation during force field learning. *Exp. brain Res.* (2017). doi:10.1007/s00221-017-4997-1

REVIEWERS' COMMENTS:

Reviewer #1 (Remarks to the Author):

The authors have largely addressed the issues raised. One remaining concern is single unit vs multiunit. The authors cite a manuscript that is still undergoing peer-review. In general, I have little doubt that a given 'manifold' can be estimated with multiunit and single unit. However, it seems a very different statement to claim what manifold is shared between two tasks. I could imagine qualitatively and quantitatively different results if only single units (e.g. lower firing rates at baseline with more variable task-related increases) vs poorly isolated multi-units are used (i.e. will likely have higher firing rates and perhaps more consistent firing as it is already a 'population' sample). Moreover, it is reasonable to hypothesize that single units may even be fully task specific while multiunits may be less likely to demonstrate such a property. At the very least, this is worth adding to the discussion.

Reviewer #2 (Remarks to the Author):

I commend the authors for a detailed and thorough revision of their manuscript. This paper has substantially improved during the revision process.

I thank the reviewers for addressing all of the major concerns (MaCs) in my initial review.

MaC1: Using the tensor maximum entropy method to generate control data is very satisfactory and a definite improvement over methods previously employed in the manuscript.

MaC2: It is indeed unfortunate that a comparison was not possible. However, I think the main results of this paper are still useful and novel despite this.

MaC3: This discussion indeed improves the paper. I was also referring to the topics now included in the introduction, paragraph 4.

Moderate concerns (MoCs):

MoC1: This is an interesting point that it may be worth including a paragraph about in the results, with the corresponding figures in the supplementary material. In general, showing that the number of neural modes to explain a high amount of variance does not linearly depend on the dimensionality of the EMG data or the inherent complexity of the task, if true, would be a useful addition to the paper.

MoC4: The logical argument has indeed improved.

MoC5: The authors do indeed show a major target-related neural mode, which is precisely useful to predict the target-related EMG activity. This point has been relegated to the supplementary material in the revision, although I believe that it warrants some discussion in the main text. Specifically, it is an important point that the R^2 values of the prediction in Fig. 6B largely comes from the accuracy in predicting the task-related EMG as shown in Supp. Fig. 8D, coupled with the high VAF of the target-related latent EMG activity. Please also mention the total R^2 of decoding the EMG from all 12 neural modes in the results section of the main text, which is currently in the methods section.

MoC6: I thank the reviewers for including this information in the revision.

MoC7: Ok

Minor Concerns (MiCs): all ok. Thank you for the detailed response to MiC8. Additionally, Supp. Fig. 5B is not actually necessary, I apologize for having suggested it in MiC7. Feel free

to remove it.

Additional Comments:

- A) The term 'manifold view' or 'viewed from the neural manifold' are used in the manuscript. Although described in the introduction, this term is not intuitively clear – consider putting quotation marks around the term 'manifold view' when introducing it. 'Viewed from the neural manifold' is especially quite unclear in the title.**
- B) Please specify in the abstract that the 'task-independent mapping onto muscle activity' is actually 'target-dependent' (which is not wholly independent of task).**
- C) Figure 1C is still unclear. What is labeled as 'Activity (u1)' is actually the 'time - dependent activation of the neural mode u1', as mentioned in the legend. Please relabel 'Activity (u1)' and 'Activity (u2)'. Additionally, it may be useful to draw the principal angles in Fig. 1D.**
- D) Please indicate the values for the mean and s.d. in Fig. 2(i), either in the figure or the caption.**
- E) It is a little hard to see the light grey circles as in Fig. 4C. Consider making them darker.**
- F) Did all the tasks in a session take roughly around the same time? Did you rescale the task time for the dPCA analysis? If so, how? (truncation is mentioned in the methods – but are you not cutting out significant portions of the trial for many trials?)**
- G) Please report how non-orthogonal the different dPC neural modes are.**
- H) It seems like the task-dependent activity begins before the presentation of the target – is this true? This may reflect the structure of task presentation.**
- I) Please specify how many total EMG channels (dimensions) there were for each monkey.**
- J) Please add back the significance tests as performed in Fig. 3A for Supp. Fig. 3E-F.**

Reviewer #3 (Remarks to the Author):

The authors have done a responsible job with responding to the reviewer comments. I have only a few remaining comments:

- 1) The PCA analysis is performed on single-trial activity, whereas the dPCA analysis is performed on the trial-averaged activity (PSTHs). Because the single-trial activity contains both the PSTHs and trial-to-trial variability, it makes the PCA analysis more difficult to interpret. Can the authors justify why they did the PCA analysis based on single-trial activity rather than PSTHs? To what extent are the results shown in Fig 3 driven by the PSTHs relative to being driven by the trial-to-trial variability?**
- 2) For Fig 5e, the authors state that the "simulated activity captured the covariance structure of the actual neural data." However it looks like the curves for the model data are quite different from the curves for the actual data. This claim should be substantiated quantitatively.**
- 3) The Russo et al. 2018 paper (which came out after the first round of review of the current manuscript) is highly relevant. Are the results in the last section of Results ("From neural modes to muscle commands") consistent with those in Russo et al.? To what extent is the neural activity related to EMGs the activity that is orthogonal to the "record player" (using the analogy from Russo et al.)?**
- 4) In the neuron dropping analysis (Fig. 3e), it is unclear how principal angles can be**

measured if the two sets of selected neurons are not identical. Measuring principal angles requires that the modes reside in the same high-d space.

Response to Reviewers' comments

We are glad for the reviewers' appreciation of our efforts to address their original concerns about the paper. Below follow the responses to their second round of comments on our manuscript. We attach a revised version of the paper, with the changes highlighted in red, after these responses.

Reviewer #1

The authors have largely addressed the issues raised. One remaining concern is single unit vs multiunit. The authors cite a manuscript that is still undergoing peer-review. In general, I have little doubt that a given 'manifold' can be estimated with multiunit and single unit. However, it seems a very different statement to claim what manifold is shared between two tasks. I could imagine qualitatively and quantitatively different results if only single units (e.g. lower firing rates at baseline with more variable task-related increases) vs poorly isolated multi-units are used (i.e. will likely have higher firing rates and perhaps more consistent firing as it is already a 'population' sample). Moreover, it is reasonable to hypothesize that single units may even be fully task specific while multiunits may be less likely to demonstrate such a property. At the very least, this is worth adding to the discussion.

We thank the reviewer for useful feedback in the first round of reviews, and are glad that the main issues have been addressed.

The reviewer raises an interesting point. One of the key properties of neural manifolds identified in several different brain areas is that the population-wide activity they capture is shared across virtually all recorded units (see the distribution of weights in Suppl. Fig. 3c,d and similar figures in (Kaufman et al., 2014; Kobak et al., 2016)). This observation implies that any sufficiently large sample of single units or multi-units would lead to the same manifold and the same latent activity within the manifold (Gao et al., 2017; Gao and Ganguli, 2015). Our neuron dropping analysis in Suppl. Fig. 3e,f provides direct evidence that the orientation of the manifold and its latent activity are not affected by the precise identity of the units used in the analysis. Moreover, Trautmann et al showed that using single units or multi-units does not affect either of these properties (Trautmann et al., 2017). All this evidence supports our claim that population-wide features, such as manifold orientation within the high-dimensional neural space and latent activity within the manifold, cannot be affected by using multi-units rather than single units.

For completeness, and to dispel any remaining doubts, we provide a direct comparison for the example wrist dataset used in Fig. 3. We used sorted units ($n=64$; see a few example single units in Fig. R1a) instead of $n=68$ multi-units used to obtain Fig. 3a for the wrist tasks. As for Fig. 3a, we obtained the 12-dimensional task-specific neural manifolds for the four wrist tasks using PCA, and used principal angles for across-task pairwise comparisons on manifold orientation. Note the remarkable similarity between the results shown in Fig. R1b and those in Fig. 3a in the paper; the latter were based on multi-units. These results indicate that task-specific manifolds identified using only sorted neurons are also similarly oriented in the high-dimensional neural space. The covariance structure of single unit activity is thus also preserved across wrist tasks. These results appear as a new panel, Suppl. Fig. 4e.

Figure R1. Using single units or multi-units to compute the neural manifold yields similar results. **(a)** ISI distributions (left) and average action potential waveforms (right) for four example sorted units for the wrist dataset used in Fig. 3a. Each task is shown in a different color (legend). Scale bars: ISI, 1 % probability; action potential waveforms: horizontal, 100 μs; vertical, 100 μV. **(b)** Principal angles for the same session of wrist tasks from Monkey J shown in paper Fig. 3a. Each pairwise comparison is shown as one colored trace (see legend). Leading principal angles were far below the $P < 0.001$ significance level (dashed gray line), indicating significant similarities in the structure of the neural modes across tasks.

Reviewer #2

I commend the authors for a detailed and thorough revision of their manuscript. This paper has substantially improved during the revision process.

We are glad that the reviewer is satisfied. We agree that the paper has improved substantially both in terms of control analyses and clarity of presentation, and thank the reviewer for their earlier set of comments.

I thank the authors for addressing all of the major concerns (MaCs) in my initial review.

MaC1: Using the tensor maximum entropy method to generate control data is very satisfactory and a definite improvement over methods previously employed in the manuscript.

MaC2: It is indeed unfortunate that a comparison was not possible. However, I think the main results of this paper are still useful and novel despite this.

MaC3: This discussion indeed improves the paper. I was also referring to the topics now included in the introduction, paragraph 4.

Moderate concerns (MoCs):

MoC1: This is an interesting point that it may be worth including a paragraph about in the results, with the corresponding figures in the supplementary material. In general, showing that the number of neural modes to explain a high amount of variance does not linearly depend on the dimensionality of the EMG data or the inherent complexity of the task, if true, would be a useful addition to the paper.

We agree with the reviewer that this is an interesting point, one that we intend to explore in future work by using datasets for a wider variety of behaviors. Unfortunately, all the additions needed to address the comments from all three reviewers have substantially increased the length of the paper. We have been asked by the Editor to shorten the manuscript, which prevents us from adding new material at this point.

MoC4: The logical argument has indeed improved.

MoC5: The authors do indeed show a major target-related neural mode, which is precisely useful to predict the target-related EMG activity. This point has been relegated to the supplementary material in the revision, although I believe that it warrants some discussion in the main text. Specifically, it is an important point that the R^2 values of the prediction in Fig. 6B largely comes from the accuracy in predicting the task-related EMG as shown in Supp. Fig. 8D, coupled with the high VAF of the target-related latent EMG activity. Please also mention the total R^2 of decoding the EMG from all 12 neural modes in the results section of the main text, which is currently in the methods section.

We had relegated these results to the Supplement because we wanted to keep the central results as simple as possible, given the complexity of the paper and the many different analyses we report. However, we agree that it is an important observation, so we have now included one new sentence in the Results. Also, following the reviewers' request, we now report the actual R^2 in the caption to Fig. 6b.

MoC6: I thank the authors for including this information in the revision.

MoC7: Ok

Minor Concerns (MiCs): all ok. Thank you for the detailed response to MiC8. Additionally, Supp. Fig. 5B is not actually necessary, I apologize for having suggested it in MiC7. Feel free to remove it.

We appreciate the reviewer's comment about Supp. Fig. 5B. However, the other reviewers also asked us to show that the main results hold for different dimensionalities, so the supplementary figure remains.

Additional Comments:

A) The term 'manifold view' or 'viewed from the neural manifold' are used in the manuscript. Although described in the introduction, this term is not intuitively clear – consider putting quotation marks around the term 'manifold view' when introducing it. 'Viewed from the neural manifold' is especially quite unclear in the title.

The expression *manifold view* now appears in italics in the Introduction; Nature Communications does not allow for quotation marks. One important goal of our paper is to promote the *manifold view*: the view that neural function is built on the activation of the neural modes rather than on the independent modulation of single units. It is this framework that allowed us to analyze data from different tasks and provide quantitative comparisons that would not otherwise be available.

Since the reviewer finds the term unclear but not incorrect, we request that we be allowed to retain the originally proposed title. We feel it reflects an important aspect of our conception of the work. Also, as the *manifold view* becomes more prevalent, as we expect and hope it will, an increasing familiarity with the concept will make it more intuitively clear.

B) Please specify in the abstract that the 'task-independent mapping onto muscle activity' is actually 'target-dependent' (which is not wholly independent of task).

Done

C) Figure 1C is still unclear. What is labeled as 'Activity (u1)' is actually the 'time-dependent activation of the neural mode u1', as mentioned in the legend. Please relabel 'Activity (u1)' and 'Activity (u2)'. Additionally, it may be useful to draw the principal angles in Fig. 1D.

Done. We have replaced the 'Activity (u1)' and 'Activity (u2)' labels by ' L_1 ' and ' L_2 ', and modified the figure caption accordingly. The principal angles have been described in the text where Fig. 1D is discussed.

D) Please indicate the values for the mean and s.d. in Fig. 2(i), either in the figure or the caption.

Added to the caption.

E) It is a little hard to see the light grey circles as in Fig. 4C. Consider making them darker.

Done.

F) Did all the tasks in a session take roughly around the same time? Did you rescale the task time for the dPCA analysis? If so, how? (truncation is mentioned in the methods – but are you not cutting out significant portions of the trial for many trials?)

The movement time for these tasks was very similar; e.g., for the example session in Fig. 4 the duration was: 845 ± 55 ms across all trials. We truncated the time courses to the duration of the fastest trial within the session (780 ms); the deleted portions were short and not significant to our goal of understanding movement generation. We have clarified the procedure in Methods, Page 21.

G) Please report how non-orthogonal the different dPC neural modes are.

Most pairs of dPC modes were close to orthogonal, as indicated by the distribution of pairwise angles between dPCs in high-dimensional neural space (Figure R2). We have added this figure as a new panel, Suppl. Fig. 6g.

Figure R2. Distribution of pairwise angles between dPC neural modes. Most of these angles were close to orthogonal (mean \pm s.d., $73.2 \pm 13.1^\circ$). Data pooled across all monkeys, wrist sessions, tasks, and dPC mode comparisons.

H) It seems like the task-dependent activity begins before the presentation of the target – is this true? This may reflect the structure of task presentation.

Yes, some task-dependent activity does seem to begin before target presentation. Our interpretation of this interesting result is that since the tasks were performed in blocks, the monkeys could recognize the task – even though the order of the tasks was randomized – and adjust their motor plan accordingly. We had already included a statement in this regard in the dPCA section of the results; see paragraph beginning with ‘The third row in Fig. 4b ...’.

I) Please specify how many total EMG channels (dimensions) there were for each monkey.

We apologize for having overlooked this. We recorded from different numbers of muscles depending on the session and the monkey; this number ranged from 5 to 12 (mean \pm s.d., 8.2 ± 2.8). This is now reported towards the end of the ‘Experimental subjects’ section of the Methods.

J) Please add back the significance tests as performed in Fig. 3A for Supp. Fig. 3E-F.

Done.

Reviewer #3

The authors have done a responsible job with responding to the reviewer comments. I have only a few remaining comments:

We thank the reviewer for insightful comments that improved the quality of our paper, and we are pleased that our revision is satisfactory.

1) The PCA analysis is performed on single-trial activity, whereas the dPCA analysis is performed on the trial-averaged activity (PSTHs). Because the single-trial activity contains both the PSTHs and trial-to-trial variability, it makes the PCA analysis more difficult to interpret. Can the authors justify why they did the PCA analysis based on single-trial activity rather than PSTHs? To what extent are the results shown in Fig 3 driven by the PSTHs relative to being driven by the trial-to-trial variability?

We decided to use single trials because we want to include as much behavioral variability as possible. All the analyses reported in the paper are based on single trials, except for dPCA, which requires trial averaging. Trial averaging results in a large reduction in the number of data points available for dimensionality reduction algorithms. So, while we implemented dPCA for the combination of four wrist tasks with six targets each, the method could not be implemented for the grip task (two or three targets, depending on the monkey) or ball task (one target). Consider trial averaging for either of these two tasks – we would end up with only two or three latent trajectories for the grip task and one for the ball task. If we sample neural activity in 20 ms bins during a 1 s long window, we would end up with 100-150 data points for the grip task and only 50 data points for the ball task. We think that it would be challenging to obtain a reliable manifold with so few points in a neural space of 50 to 100 dimensions.

Nevertheless, it is certainly feasible to perform PCA and obtain manifolds using trial-averaged data for the wrist tasks; they have six or eight targets, and trial averaging still provides enough data samples in the high-dimensional neural space. We have repeated the principal angle analysis shown in Fig. 3 but now using trial-averaged data, in order to address the reviewer's comment. We computed the task specific 12-dimensional manifolds, and performed across-task pairwise comparisons on manifold orientation using principal angles. As in Fig. 3b, we counted the number of neural modes for which all principal angles were significantly small ($P < 0.001$). The results, shown below in Fig. R3, are remarkably similar to those in Fig. 3b in the paper. We conclude that the covariance across neural units does not vary by much when computed from the PSTHs instead of single trials, as the leading principal angles are still well below the 0.001 significance threshold.

Figure R3. Neural manifolds computed based on PSTHs are well-aligned in high-dimensional space. Number of neural modes for which all principal angles were significantly small across all wrist sessions, monkeys, and task pairs ($P < 0.001$).

2) For Fig 5e, the authors state that the "simulated activity captured the covariance structure of the actual neural data." However it looks like the curves for the model data are quite different from the curves for the actual data. This claim should be substantiated quantitatively.

We apologize for the inaccuracy of our statement. The eigenvalue distribution curves are indeed different: neural modes explain considerably more variance for the model data than for the actual data (e.g., by a factor of 1.3 for 12 neural modes). This reflects the fact that much of the variance in the actual neural data does not relate to the EMGs, and thus cannot be predicted from EMGs. The dPCA-based analysis of the relation between neural and EMG activity revealed that ~70 % of the total neural variance – captured by the time-, target- and task-related neural modes – was a very poor predictor of EMG activity (Figure 6b,c). Since this large amount of neural variance cannot be generated by the model in Fig. 5d, less neural modes are needed to explain a certain amount of variance for the synthetic than for the actual data. We have corrected the text according to this; see bottom of Page 11 and beginning of Page 12.

3) The Russo et al. 2018 paper (which came out after the first round of review of the current manuscript) is highly relevant. Are the results in the last section of Results ("From neural modes to muscle commands") consistent with those in Russo et al.? To what extent is the neural activity related to EMGs the activity that is orthogonal to the "record player" (using the analogy from Russo et al.)?

We are familiar with the very insightful paper by Russo and colleagues, which we already referred to several times in the revised paper. Our results are compatible with theirs. First, we view the time-related but task- and target-independent dPCs as related to the components that “make the record spin”. A comment to that regard appears in the revised version, Discussion section, where we wrote: “Time-related neural modes may also support the generation of robust motor commands (Russo et al., 2018).” Second, similar to findings in Russo et al., the target-related neural modes that we found to be strongly predictive of EMGs were mostly orthogonal to the time-related neural modes; see Fig. R4 below for the distribution of angles between time-related and target-related dPCs.

Figure R4. Distribution of pairwise angles between time-related and target-related dPC neural modes. Most of these angles were close to orthogonal (mean \pm s.d., $73.2 \pm 13.3^\circ$). Data pooled across all monkeys, wrist sessions, tasks, and dPC mode comparisons.

4) In the neuron dropping analysis (Fig. 3e), it is unclear how principal angles can be measured if the two sets of selected neurons are not identical. Measuring principal angles requires that the modes reside in the same high-d space.

The reviewer is right that our description of the neuron dropping analyses in Suppl. Fig. 3e was not sufficiently clear; we have clarified the procedure in the ‘Control analyses’ section of the Methods, Page 27.

The procedure was as follows: we dropped a certain percentage (10 %, 20 %, 30 %, etc.) of neural units by setting their corresponding activity to zero. This was done 100 times for each percentage, always dropping the same number of units but randomly picking each time a different set of units. In all cases, the dimensionality of the data was preserved as that of the original neural space, thus allowing us to compare manifold orientations using principal angles. The same approach to neural unit dropping was used for the CCA analysis reported in Suppl. Fig 3f. Note that we have slightly modified these two figures to address point (J) from Reviewer #2 by incorporating the $P < 0.001$ significance thresholds.

References

- Gao, P., Ganguli, S., 2015. On simplicity and complexity in the brave new world of large-scale neuroscience. *Curr. Opin. Neurobiol.* 32, 148–155. doi:10.1016/j.conb.2015.04.003
- Gao, P., Trautmann, E., Yu, B.M., Santhanam, G., Ryu, S., Shenoy, K., Ganguli, S., 2017. A theory of multineuronal dimensionality, dynamics and measurement. *bioRxiv* 214262. doi:10.1101/214262
- Kaufman, M.T., Churchland, M.M., Ryu, S.I., Shenoy, K. V, 2014. Cortical activity in the null space: permitting preparation without movement. *Nat. Neurosci.* 17, 440–8. doi:10.1038/nn.3643
- Kobak, D., Brendel, W., Constantinidis, C., Feierstein, C.E., Kepecs, A., Mainen, Z.F., Qi, X., Romo, R., Uchida, N.,

Machens, C.K., 2016. Demixed principal component analysis of neural population data. *Elife* 5, 1–37. doi:10.7554/eLife.10989

Russo, A.A., Bittner, S.R., Perkins, S.M., Seely, J.S., London, B.M., Lara, A.H., Miri, A., Marshall, N.J., Kohn, A., Jessell, T.M., Abbott, L.F., Cunningham, J.P., Churchland, M.M., 2018. Motor Cortex Embeds Muscle-like Commands in an Untangled Population Response. *Neuron* 1–14. doi:10.1016/j.neuron.2018.01.004

Trautmann, E.M., Stavisky, S.D., Lahiri, S., Ames, K.C., Kaufman, M.T., Ryu, S.I., Ganguli, S., Shenoy, K. V., 2017. Accurate estimation of neural population dynamics without spike sorting 1–42. doi:10.1101/229252